# P-Glycoprotein Inhibitors Differently Affect *Toxoplasma gondii*, *Neospora caninum* and *Besnoitia besnoiti* Proliferation in Bovine Primary Endothelial Cells

**DOI:** 10.3390/pathogens10040395

**Published:** 2021-03-25

**Authors:** Camilo Larrazabal, Liliana M. R. Silva, Learta Pervizaj-Oruqaj, Susanne Herold, Carlos Hermosilla, Anja Taubert

**Affiliations:** 1Biomedical Research Center Seltersberg, Institute of Parasitology, Justus Liebig University Giessen, 35392 Giessen, Germany; Liliana.Silva@vetmed.uni-giessen.de (L.M.R.S.); Carlos.R.Hermosilla@vetmed.uni-giessen.de (C.H.); Anja.Taubert@vetmed.uni-giessen.de (A.T.); 2The Cardio-Pulmonary Institute (CPI), 35392 Giessen, Germany; Learta.Pervizaj-Oruqaj@innere.med.uni-giessen.de (L.P.-O.); Susanne.Herold@innere.med.uni-giessen.de (S.H.); 3Member of the German Center for Lung Research (DZL), Department of Pulmonary and Critical Care Medicine and Infectious Diseases, Universities of Giessen and Marburg Lung Center (UGMLC), Justus-Liebig University Giessen, 35392 Giessen, Germany

**Keywords:** P-glycoprotein, ABCB1-transporter, *Toxoplasma gondii*, *Neospora caninum*, *Besnoitia besnoiti*, verapamil, valspodar, tariquidar

## Abstract

Apicomplexan parasites are obligatory intracellular protozoa. In the case of *Toxoplasma gondii*, *Neospora caninum* or *Besnoitia besnoiti*, to ensure proper tachyzoite production, they need nutrients and cell building blocks. However, apicomplexans are auxotrophic for cholesterol, which is required for membrane biosynthesis. P-glycoprotein (P-gp) is a transmembrane transporter involved in xenobiotic efflux. However, the physiological role of P-gp in cholesterol metabolism is unclear. Here, we analyzed its impact on parasite proliferation in *T. gondii*-, *N. caninum*- and *B. besnoiti*-infected primary endothelial cells by applying different generations of P-gp inhibitors. Host cell treatment with verapamil and valspodar significantly diminished tachyzoite production in all three parasite species, whereas tariquidar treatment affected proliferation only in *B. besnoiti*. 3D-holotomographic analyses illustrated impaired meront development driven by valspodar treatment being accompanied by swollen parasitophorous vacuoles in the case of *T. gondii*. Tachyzoite and host cell pre-treatment with valspodar affected infection rates in all parasites. Flow cytometric analyses revealed verapamil treatment to induce neutral lipid accumulation. The absence of a pronounced anti-parasitic impact of tariquidar, which represents here the most selective P-gp inhibitor, suggests that the observed effects of verapamil and valspodar are associated with mechanisms independent of P-gp. Out of the three species tested here, this compound affected only *B. besnoiti* proliferation and its effect was much milder as compared to verapamil and valspodar.

## 1. Introduction

*Toxoplasma gondii*, *Neospora caninum* and *Besnoitia besnoiti* are obligatory intracellular parasitic protists known as cyst-forming coccidia and belonging to the family Sarcocystidae of the phylum Apicomplexa. Many apicomplexans are causal agents of important human and animal diseases with variable host specificity and clinical outcomes [1]. In specific, *T. gondii* has the widest known host range among eukaryotic parasites as virtually all mammals and even some birds can be infected. This species represents a serious health threat [2] as prenatal infections may lead to abortions in humans and sheep [3,4]. In contrast, the closely related coccidian *N. caninum* does not infect humans, but it is one of the most important abortive agents in cattle worldwide [5,6]. The third species, *B. besnoiti,* causes bovine besnoitiosis, which is emergent in Europe and leads to massive alterations of the skin and mucosa of cattle as well as infertility in bulls [7]. During the acute stage of infection, cyst-forming coccidian parasites undergo asexual intracellular proliferation, quickly releasing a considerable number of offspring (tachyzoites). This stage is linked with clinical manifestations of the parasite infection, and is followed by a slow replicative process with bradyzoites enclosed in cysts, which are infective for the definitive hosts (felids and canids) [1].

Considering the relevance of the tachyzoite replication in the acute phase of infection, several reports demonstrated that primary bovine umbilical vein endothelial cells (BUVEC) represent suitable host cells for in vitro replication of all three species [8,9,10,11,12], allowing fast tachyzoite replication in a setting close to the in vivo situation. During this replication process, one of the most demanded molecules is cholesterol. Being auxotrophic for cholesterol synthesis, coccidian parasites either trigger its uptake from exogenous sources or induce the de novo synthesis of this compound by infected host cells [8,13,14,15,16]. Low density lipoprotein (LDL)-mediated endocytosis represents the main uptake mechanism in many types of host cells [17]. *T. gondii*, *N. caninum* and *B. besnoiti* were reported to rely on host cell LDL endocytosis for cholesterol uptake during tachyzoite replication [8,13,14]. To prevent toxic accumulation of free cholesterol in the cell, most excess of it is esterified and stored in lipid-rich organelles, such as lipid droplets [18,19], being available for parasite consumption [8,13]. Thus, to satisfy an increased need of cholesterol and other lipids in infected host cells, lipid droplet size and number are reported to be significantly increased [8,20].

Other strategy to face the challenge of lipid imbalance is the use of ATP binding cassette (ABC) transporter-mediated efflux mechanisms [21,22]. ABC transporters are highly evolutionarily conserved and ubiquitous molecules that mediate a broad range of physiological functions [21]. In eukaryotic cells, one of the most extensively studied members of the ABC transporter family is P-glycoprotein (P-gp; syn. ABCB1) [21], the expression of which is frequently correlated with the removal of hydrophobic compounds from the cell [22,23]. This transmembrane protein, also called multidrug resistance protein 1 (MDR1), is vastly associated with drug resistance phenomena in cancer cells [24]. Noteworthy, the participation of this transporter in drug resistance has been proposed also for protozoan parasites, such as *T. gondii* [25], *Leishmania* spp. [26] and *Plasmodium falciparum* [27], as well as for helminths like *Schistosoma mansoni* [28], *Teladorsagia circumcincta*, *Haemonchus contortus* [29] and *Echinococcus granulosus* [30]. Despite the well-known role of P-gp in drug resistance development, little is known about its impact on cell metabolism. As such, this protein seems to be involved in host cellular cholesterol transport to the parasitophorous vacuole (PV) during *T. gondii* and *N. caninum* replication [31], and might be necessary to maintain cholesterol homeostasis during coccidian infections. P-gp inhibition was demonstrated to suppress replication in some coccidia and microsporidia [31,32]. However, P-gp blockers represent a heterogeneous group of compounds. By now, three generations of pharmacological P-gp blockers have been developed, differing greatly in chemical and pharmacological terms [33]. 

In this work, we explored the effects of P-gp inhibitors on in vitro proliferation of different fast-replicating coccidian species (*T. gondii*, *N. caninum* and *B. besnoiti*) by testing the L-type Ca^++^ channel blocker verapamil [34], the non-immunosuppressive cyclosporine D derivative valspodar [35,36] and the highly selective allosteric inhibitor tariquidar [37,38], as representative compounds of each P-gp blocker generation. The aim of this work was to identify and compare anti-coccidial properties of well-known compounds for future drug repurposing reasons.

## 2. Results

### 2.1. Different Generations of P-Gp Inhibitors Vary in Their Impact on Tachyzoite Replication

The efficacy of verapamil, valspodar and tariquidar treatment on tachyzoite replication was evaluated via functional inhibition assays, determining the number of freshly released tachyzoites present in the medium at 48 h post infection (p. i.). The current data showed that verapamil effectively inhibited parasite replication in a dose-dependent manner (Figure 1). Thus, *T. gondii* replication (Figure 1A) was diminished by 45.7 ± 11.7% (*p* = 0.0508) and 84.04 ± 7.6% (*p* = 0.001) at 20 and 40 µM, respectively. At the same concentrations, this compound led to a reduction of *N. caninum* replication (Figure 1B) by 64.9 ± 10.4% (*p* = 0.009) and 84.5 ± 2.7% (*p* = 0.0001), respectively, and to a decrease of *B. besnoiti* parasite production (Figure 1C) by 36.6 ± 5.9% (*p* = 0.07) and 84.5 ± 2.7 (*p* = 0.0005). Lower concentrations of verapamil (5 and 10 µM) failed to significantly affect tachyzoite proliferation in any species studied here.

Alike verapamil, treatment with valspodar also caused a dose-dependent reduction of tachyzoite proliferation (Figure 2). Thus, *T. gondii* replication (Figure 2A) was reduced by 67.8 ± 10.4% (*p* = 0.78), 91.6 ± 3.4% (*p* = 0.07) and 98.9 ± 0.3% (*p* = 0.002) at 1.25, 2.5 and 5 µM concentration, respectively. Likewise, *N. caninum* replication (Figure 2B) was reduced at 2.5 and 5 µM concentrations, leading to a proliferation diminishment of 64.0 ± 5.0% (*p* = 0.0068) and 92.8 ± 2.5% (*p* = 0.0001), respectively. Furthermore, *B. besnoiti* replication (Figure 2C) was blocked at 1.25, 2.5 and 5 µM valspodar by 37.94 ± 18.3% (*p* = 0.9), 94.6 ± 2.5% (*p* = 0.09) and 99.6 ± 0.3% (*p* = 0.003), respectively.

In contrast to valspodar and verapamil, tariquidar treatment had differential impact on tachyzoite replication (Figure 3) by exhibiting inhibitory efficacy against *B. besnoiti* proliferation at 1 and 2 µM, showing a reduction of 35.9 ± 15.2%, (*p* = 0.023) and 51.7 ±14.8 (*p* = 0.002) of tachyzoite replication, respectively, but failing to block either *T. gondii* or *N. caninum* proliferation.

### 2.2. Infection Rates Are Differentially Influenced by Different P-Gp Inhibitors

Host cell infection is the earliest step necessary for successful parasite proliferation and progression. To assess direct effects of inhibitors on tachyzoite stages, we first treated live tachyzoites and then evaluated their invasive capacities for BUVEC (Figure 4, tachyzoite treatment). Non-treated tachyzoites infected 47.6% ± 1.2, 62.0 ±2.3 and 55.1 ± 6.6 of the host cells in the case of *T. gondii*, *N. caninum* and *B. besnoiti*, respectively. Tachyzoite infectivity was not affected by verapamil (40 µM) treatment. In contrast, valspodar (5 µM) treatment resulted in a decrease of infection rates with *T. gondii*, *N. caninum*, and *B. besnoiti* by 7.8% (*p* = 0.009), 14.13% (*p* = 0.013) and 19.3% (*p* = 0.022), respectively. Tariquidar tachyzoite treatment (2 µM) affected exclusively *B. besnoiti* infectivity, thereby reducing the infection rate by 16.41% (*p* = 0.03). 

Host cell pre-treatments were performed to assess inhibitor-related effects on its permissiveness for parasite invasion (Figure 4, host cell treatment). In non-treated host cells, 42.2 ± 2.7%, 59.0 ± 2.7% and 53.7 ± 1.8% of the host cells were found infected with *T. gondii*, *N. caninum* and *B. besnoiti* tachyzoites, respectively. Host cell treatment with verapamil (40 µM) and tariquidar (2 µM) had no effect on permissiveness as judged by comparable infection rates at 4 h p. i. In contrast, host cell pre-treatment with valspodar (5 µM) led to reduced infection rates with all three parasites (reduction of 20.12% (*p* = 0.036), 6.35% (*p* = 0.085) and 9.7% (*p* = 0.008) in the case of *T. gondii*, *N. caninum* and *B. besnoiti*, respectively).

### 2.3. P-Gp Inhibitor-Driven Morphological Alterations in Primary Bovine Endothelial Cells

To assess the impact of inhibitor treatments on host cell physiology, BUVEC morphology was analyzed via live cell 3D-holotomographic microscopy. In uninfected BUVEC cells, 48 h verapamil treatment induced a considerable accumulation of the dense globular structures (refractive index of 1.3488 ± 0.0048) in the cytoplasm and their displacement from the center to the periphery of the cells (Appendix A). As for infected cells, this effect could clearly be observed only in the case of *N. caninum*, whilst for the two other coccidian species the results were inconclusive (Figure 5). A change of meront morphology was not observed in verapamil-treated host cells. Given that the globular structures resembled cytoplasmic lipid droplets, we here additionally performed experiments applying BODIPY 493/503, which is a well-accepted probe for neutral lipids and often used for lipid droplet detection. When using BODIPY 493/503 staining in live cell 3D-holotomography, identical globular structures were marked by bright green fluorescence (Figure 6A) as observed before, and these structures were much less evident in vehicle-treated control cells (Figure 6A). Moreover, the same observation was made in infected host cells (Appendix A). Given that neutral lipids are typically stored in lipid droplets, these data indicated a verapamil-driven increase of lipid droplet formation in BUVEC. Quantitative flow cytometric analyses confirmed a verapamil-induced increase in BODIPY 493/503-driven signals, whilst valspodar and tariquidar treatment had no effect on lipid droplet formation (Figure 6B). Thus, treatment with verapamil led to a 74.1% increase of BODIPY-derived mean fluorescence intensity when compared to control conditions (*p* = 0.026)

In contrast to verapamil-treated cells, valspodar- and tariquidar-treated BUVEC showed normal cytoplasmic morphology lacking any vesicular accumulation. In the case of tariquidar, no effects on host cell or meront morphology could be detected in any parasite species tested here (data not shown). In contrast, valspodar treatment led to impaired meront development in all parasite species studied here (Figure 7). Thus, a considerable decrease in meront sizes was observed (Figure 7A). Specifically, valspodar treatment reduced the meront diameter by 45.2%, 40.2% and 29. 2% for *T. gondii*, *N. caninum* and *B. besnoiti* (*p* < 0.0001 for all three species) (Appendix A). Interestingly, exclusively in the case of *T. gondii* infections, valspodar treatment caused alterations in some of the PV as these parasitic structures appeared swollen. Moreover, parasite replication was consistently arrested at one-tachyzoite-stage/PV as detected throughout the entire experiment and still observed at 36 and 46 h p. i. (Figure 7B, arrows).

### 2.4. P-Gp Inhibitor Treatment Does Not Cause Cytotoxic Damage to Host Cells or Tachyzoites 

To determine if the treatment with P-gp inhibitors evoke a cytotoxic effect on endothelial host cells or tachyzoites cytotoxicity assays were performed. As illustrated in Appendix A, treatments with verapamil (40 µM), valspodar (5 µM) and tariquidar (2 µM) did not induce significant colorimetric changes in the formazan product compared to the vehicle control (DMSO 0.01%). Similarly, the trypan blue exclusion test showed an average viability of 88.9 ± 3.8% for *T. gondii*, *N. caninum* and *B. besnoiti* treated for 1 h with vehicle control (DMSO 0.01%) without significant effects provoked by verapamil (40 µM), valspodar (5 µM) or tariquidar (2 µM) treatments (Appendix A–D).

## 3. Discussion

The cyst-forming coccidia studied here, i.e., *T. gondii*, *N. caninum* and *B. besnoiti*, represent fast-replicating parasites that share the capacity of massive offspring production within 1–3 days after infection in vivo. Consequently, they all require significant amounts of building blocks to ensure successful parasite proliferation. Being auxotrophic for cholesterol, they strongly depend on the availability and efficient uptake of this nutrient for the successful proliferation [39]. Hence, P-gp implicated in the cholesterol uptake represents a promising anti-coccidial target. Here we investigated the effects of different P-gp inhibitors on the three above-mentioned parasite species. We found that verapamil treatment diminished intracellular parasite proliferation in a dose-dependent manner without cytotoxic effects on the host cells nor tachyzoites. At 40 µM, verapamil significantly reduced the replication of *T. gondii*, *N. caninum* and *B. besnoiti* tachyzoites in BUVEC by an average of 84 ± 0.8%. Comparable effects of verapamil (at concentrations of 10–100 μM) were already reported in *T. gondii* infections of mouse embryonic fibroblasts and enterocytes [40,41]. In addition, anti-parasitic effects were also documented for *P. falciparum* erythrocytic stages [42,43]. Here, we demonstrated that verapamil treatment of tachyzoites and host cells failed to affect infection rates, indicating that its anti-parasitic effect seems exclusively associated with parasite division processes, but not with active tachyzoite invasion or PV formation. In line with this, live cell 3D holotomography revealed no morphological changes of newly formed PVs and tachyzoites but indicated an accumulation of dense vesicles surrounding the PV. Thus, the anti-coccidial effect of verapamil might also be linked to intracellular lipid transport mechanisms. Interestingly, for other primary cell cultures, the impact of verapamil on cellular proliferation has been previously demonstrated to depend on calcium-dependent mechanisms [44,45,46]. Considering the importance of calcium homeostasis for tachyzoite proliferation [47], it is likely that verapamil-triggered effects were driven by a calcium-mediated pathway in the current endothelium system. However, additional mechanisms may play a role as verapamil also interacts with other key channels and transporters, such as the glucose transporter GLUT1, which mediates constitutive glucose uptake in various cell types, including endothelial cells [48]. In fibroblasts, verapamil blocks GLUT1-mediated glucose transport at both basal and stress-induced conditions [49]. Thus, glucose-related effects may have contributed to the anti-proliferative impact of verapamil reported in this study, however further experiments are necessary to address this possibility. 

Besides verapamil, we analyzed anti-coccidial effects of one of the most promising second-generation P-gp inhibitors, namely valspodar [35,36]. Current data showed that valspodar treatment significantly blocked *T. gondii*, *N. caninum* and *B. besnoiti* tachyzoite replication in infected BUVEC, without affecting the host cell and tachyzoites viability. Thus, 5 μM valspodar—depending on the studied species—reduced tachyzoite production by 92.8 ± 2.5–99.6 ± 0.3%, with an effective inhibitory concentration being eight times smaller than that of verapamil. In accordance with this, live cell 3D holotomography illustrated a marked reduction in meront sizes. In *T. gondii*, 3D-holotomographic microscopy also revealed an arrest at the single tachyzoite-stage. The mechanism underlying species-specific sensitivity is still unclear. However, anti-proliferative effects of valspodar were already demonstrated for *T. gondii* tachyzoites in permanent cell lines (Vero cells; [50]), for *P. falciparum* erythrocytic stages [51] and for *Cryptosporidium parvum*-infected Caco-2 cells [52], suggesting that valspodar-mediated effects are conserved among these apicomplexans. Given that valspodar is a derivative of cyclosporine, for which anti-parasitic activity against the same three species has been demonstrated [50,51,52], these two compounds may have common effects. Here, we furthermore documented that valspodar treatment led to altered tachyzoite infectivity in the three coccidian species studied here without any significant effect on tachyzoite viability, which is in agreement with previous data on *T. gondii* (10 µM; [25]). More interestingly, current data showed that host cell pre-treatment with this compound led to a moderate reduction of *T. gondii* and *B. besnoiti* infection rates, suggesting that valspodar additionally reduces host cell permissiveness, thereby hampering parasite invasion, even though it is an active process mainly driven by tachyzoites [53]. However, treatment with the endocytosis inhibitor dynasore also impairs *T. gondii* infectivity, proving that both tachyzoite- and host cell-derived actions are crucial for invasion [54]. Overall, the current data evidence that valspodar affected several aspects of parasite infection, i.e., tachyzoite infectivity, host cell permissiveness and tachyzoite division. 

Considering that the promiscuity associated with derivative drug design strategies was greatly overcome for P-gp third-generation inhibitors, we also evaluated effects of the selective P-gp blocker tariquidar. Tariquidar is a potent P-gp-specific allosteric inhibitor with an average IC_50_ of 50 nM [28,37,38]. Overall, tariquidar treatments were not toxic for the host cell or tachyzoites and concentrations of 1–2 µM showed a moderate effect on *B. besnoiti* proliferation but failed to influence *T. gondii* and *N. caninum* intracellular replication. Interestingly, tariquidar tachyzoite pre-treatment also affected *B. besnoiti* host-cell infectivity, thereby emphasizing species-specific effects. So far, little is known on the impact of third-generation P-gp inhibitors on apicomplexan parasite proliferation. In line with the current data, it has been previously reported that elacridar (10 μM) treatment reduced *T. gondii* proliferation and, by affecting Ca^++^ homeostasis of tachyzoites, led to hypermotility and untimely microneme secretion [40]. Thus, the current tariquidar-mediated effects on *B. besnoiti* tachyzoites may rely on a species-specific reduction of infectivity blockage and replication. 

Until now, the role of P-gp in cholesterol homeostasis is still under debate [23]. Whilst one report argues that P-gp expression does not play a major role in cholesterol homeostasis in P-gp-inducible cells, another evidenced a physiological role of P-gp in intracellular cholesterol trafficking in *T. gondii*-infected fibroblasts [31,55]. However, the fact that different studies were performed on different types of host cells can explain such oppositional findings. Interestingly, P-gp inhibitors have shown pharmacological activities beyond P-gp. Some of these compounds can also interact with other transporters affecting cellular cholesterol homeostasis [56,57]. In this context, we aimed to find out if anti-coccidial effects could be related to cholesterol homeostasis-related mechanisms. Therefore, we evaluated if inhibitor treatments affected neutral lipid accumulation. Indeed, verapamil treatment resulted in neutral lipid accumulation in BUVEC as measured by BODIPY-derived signals, whilst valspodar and tariquidar treatment did not have such an effect. So far, analyses of neutral lipid accumulation driven by P-gp are limited as some dyes may act as P-gp substrates and be actively exported from the cell [33]. To avoid data misinterpretation, we also confirmed neutral lipid accumulation via live cell 3D-holotomography. Co-localization of the RI and BODIPY fluorescence in vesicles confirmed verapamil as an inducer of neutral lipid accumulation in BUVEC. In line with this, verapamil-induced cholesterol accumulation was also reported in rabbit aortic smooth cells [46] and mouse myocardium [58].

In summary, as shown in Table 1 here we demonstrated that verapamil and valspodar treatments inhibited intracellular *T. gondii*, *N. caninum* and *B. besnoiti* tachyzoite replication in BUVEC and possess significant differences in anti-coccidial and cholesterol-related side effects. We assume that the high efficacy of valspodar is based on its effects on both infectivity and replication, and is independent of cholesterol-related pathways. In contrast, treatment with tariquidar revealed species-specific effects and led to reduced tachyzoite infectivity and proliferation only in *B. besnoiti*.

## 4. Materials and Methods

### 4.1. Host Cell Culture

Primary bovine umbilical vein endothelial cells (BUVEC) were isolated as described elsewhere [9,10]. BUVEC were cultured at 37 °C and 5% CO_2_ atmosphere in modified ECGM (modECGM) medium, by diluting ECGM medium (Promocell, Heidelberg, Germany) with M199 (Sigma-Aldrich, Munich, Germany) at a ratio of 1:3, supplemented with 500 U/mL penicillin (Sigma-Aldrich), 50 μg/mL streptomycin (Sigma-Aldrich) and 5% FCS (fetal calf serum; Biochrom, Cambridge, UK). Only BUVEC of less than three passages were used in this study.

### 4.2. Parasite Cultures

*T. gondii* (strain RH) and *N. caninum* (strain NC-1) were maintained in vitro in permanent African green monkey kidney epithelial cells (MARC145) in DMEM (Sigma-Aldrich) as described elsewhere [12]. *B. besnoiti* (strain Bb Evora04) was propagated in Madin-Darby bovine kidney cells (MDBK) [11,59] in RPMI medium (Sigma-Aldrich). All cell culture media were supplemented with 500 U/mL penicillin, 50 μg/mL streptomycin (Sigma-Aldrich) and 5% FCS (Gibco^TM^). Infected and non-infected cells were cultured at 37 °C and 5% CO_2_ atmosphere. Live tachyzoites were collected from supernatants of infected host cells (400× *g*; 10 min) and re-suspended in modECGM for further experiments. 

### 4.3. Inhibitor Treatment

BUVEC (*n* = 5) were cultured in 12-well plates (Sarstedt, Nümbrecht Germany) previously coated with fibronectin (1:400; Sigma-Aldrich). Verapamil (Cayman Chemical, Ann Arbor, MI, USA), valspodar and tariquidar (both Sigma-Aldrich) were solved in DMSO (dimethyl sulfoxide; Sigma-Aldrich) and diluted in modECGM. Inhibitor treatments were performed by supplementation with verapamil (5–40 µM), valspodar (0.6–5 µM) or tariquidar (0.2–2 µM) to fully confluent cell layers 48 h prior to infection. ModECGM with DMSO (0.01%) served as control medium. After 48 h of treatment, the inhibitors were removed by washing with plain medium. The cells were incubated with tachyzoites of *T. gondii*, *B. besnoiti* or *N. caninum* at a multiplicity of infection of 5 for 4 h. This was followed by the removal of remaining extracellular tachyzoites by washing with plain medium and inhibitor re-administration. At 48 h post infection (p. i.), tachyzoites present in cell culture supernatants were collected (800× *g*; 5 min) and counted in a Neubauer chamber. 

To estimate inhibitor effects on tachyzoite infectivity, fresh tachyzoites were treated for 1 h (37 °C, 5% CO_2_) with verapamil, valspodar or tariquidar. After washing in plain medium (800× *g*; 5 min), inhibitor-treated tachyzoites and non-treated control parasites were used for infection as described above. In addition, for host cell permissiveness assays the host cells were incubated with the inhibitors as described above (48 h pre infection) and then the inhibitors were removed and cells infected with live tachyzoites in medium without inhibitors. In both cases, at 4 h p. i., phase-contrast images (3 per experimental condition, *n* = 5) were acquired with an inverted microscope (IX81, Olympus, Tokyo, Japan) for infection rate estimation. 

### 4.4. Flow Cytometry Analysis

BUVEC (*n* = 5) were seeded into T-25 cm^2^ flasks (Sarstedt) and cultured until confluence. Thereafter, the cells were treated with verapamil (40 µM), valspodar (5 µM) or tariquidar (2 µM) for 48 h. To determine if inhibitor treatment exerted an effect on neutral lipids, pre-treated cells were stained with BODIPY 493/503 (2.5 µM, Cayman Chemical, 1 h, 37 °C, 5% CO_2_). Afterwards, cells were washed twice in PBS 1× (600× *g*; 5 min), fixed in 4% PFA (paraformaldehyde; Sigma-Aldrich) and stored at −80 °C until further analysis. The samples were analyzed by a BD LSRFortessa^®^ cell analyzer (Becton-Dickinson, Heidelberg, Germany). Cells were gated according to their size and granularity. Moreover, BODIPY 493/503-derived signals were assessed in the FL-1 channel. Data analysis was performed via FlowJo^®^ (version 10.5.0) flow cytometry analysis software (FlowJo LLC, Ashland, OR, USA).

### 4.5. Live Cell 3D-Holotomographic Microscopy

BUVEC were seeded into 35 mm tissue culture µ-dishes (Ibidi, Planegg, Germany) and cultured (37 °C, 5% CO_2_) to confluence. P-gp inhibitor treatment and parasite infections were performed as described above. Treated cell layers were placed in a top-stage incubator (Ibidi^®^) at 5% CO_2_ and 37 °C during the entire experiment. Holotomographic images were obtained via 3D cell-explorer microscope (Nanolive, Ecublens, Switzerland) equipped with a 60× magnification (λ = 520 nm, sample exposure 0.2 mW/mm^2^) and a depth of field of 30 µm. Images were analyzed using STEVE software (Nanolive) to obtain refractive index (RI)-based z-stacks. In addition, digital staining was applied according to the RI of cell organelles and intracellular tachyzoites. For neutral lipid visualization, cells were loaded with BODIPY 493/503 (2.5 µM, 1 h, 37 °C). Live cell 3D-holotomographic microscopy and analysis of BODIPY 493/503-based fluorescence were performed in parallel to prove the nature of lipid-rich organelles. Image processing was carried out by Fiji ImageJ^®^ using Z-projection and merged-channel-plugins.

### 4.6. Cell Toxicity Assays

Cell toxicity of P-gp inhibitors was assessed by colorimetric XTT tests (Promega, Madison, WI, USA) according to the manufacturer instructions. Briefly, BUVEC (*n* = 3) were cultured in 96-well plates (Greiner) and treated with verapamil (40 µM), valspodar (5 µM) or tariquidar (2 µM) in a total volume of 50 µl for 96 h. Thereafter, 50 μL of XTT working solution were added and the samples were incubated for 4 h (37 °C, 5% CO_2_ atmosphere). The resulting formazan products were estimated via optical density (OD) measurements at 590 nm and reference filter 620-nm wavelength using VarioskanTM Flash Multimode Reader (Thermo Scientific, Waltham, MA, USA). BUVEC treated with the solvent (DMSO; 0.01%) were used as negative controls.

Additionally, for experiments on parasite viability, 5 × 10^5^ tachyzoites of each parasite species were treated for 1 h with each of the studied compounds (verapamil 40 µM, valspodar 5 µM and tariquidar 2 µM; 37 °C, 5% CO_2_). Viability of tachyzoites was determined by the trypan blue (Sigma-Aldrich) exclusion staining assay [60]. Non-stained parasites were considered as viable.

### 4.7. Statistical Analysis

Statistical analyses were performed by the software GraphPad Prism^®^ 8 (version 8.4.3., www.graphpad.com) Data description was performed by presenting the arithmetic mean ± standard deviation. In addition, the non-parametric Mann–Whitney test was applied for the comparison of two experimental conditions, while Kruskal–Wallis test was used for the comparison of three or more conditions. Whenever a global comparison by the Kruskal–Wallis test indicated significance, post hoc multiple comparison was carried out using the Dunn test to compare with control conditions. The outcomes of the statistical tests were considered to indicate significant differences at *p* ≤ 0.05 (significance level).

## Figures and Tables

**Figure 1 pathogens-10-00395-f001:**
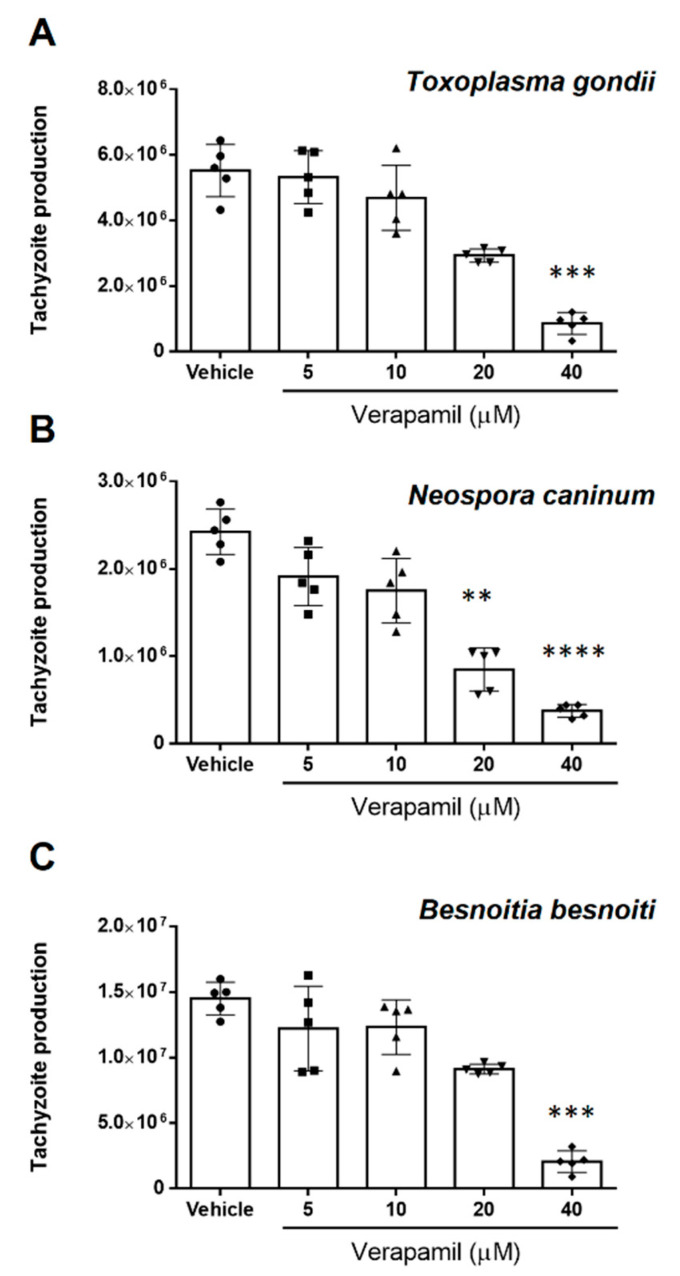
Verapamil treatment reduces *T. gondii*, *N. caninum* and *B. besnoiti* tachyzoite proliferation in a dose-dependent manner. The hosts cells pre-treated with verapamil (5, 10, 20 and 40 μM) were infected with *T. gondii* (**A**), *N. caninum* (**B**) or *B. besnoiti* (**C**) tachyzoites in inhibitor-free medium for 4 h, followed by the compound re-administration. At 48 h after infection, the number of tachyzoites present in cell culture supernatants were counted. Bars represent means of five biological replicates ± standard deviation.

**Figure 2 pathogens-10-00395-f002:**
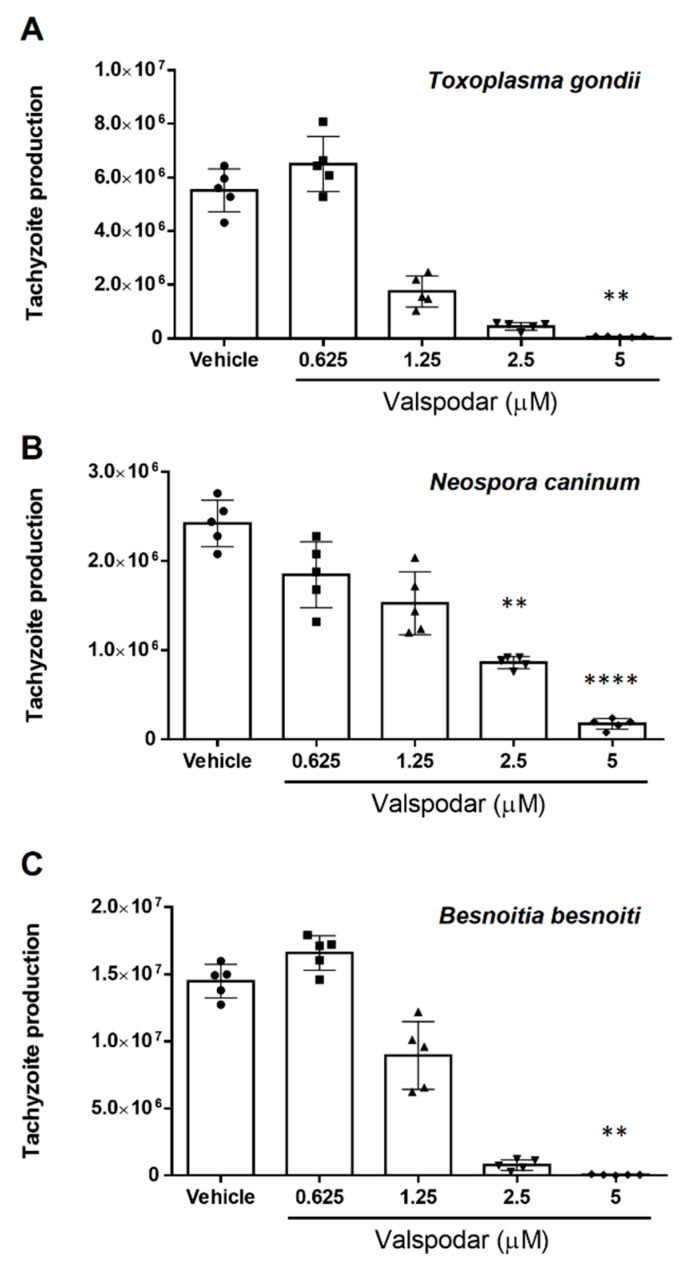
Valspodar treatment induces a dose-dependent effect on *T. gondii*, *N. caninum* and *B. besnoiti* proliferation. The host cells pre-treated with valspodar (0.625, 1.25, 2.5 and 5 µM) were infected with *T. gondii* (**A**), *N. caninum* (**B**) or *B. besnoiti* (**C**) tachyzoites in inhibitor-free medium for 4 h, followed by the compound re-administration. At 48 h after infection, the number of tachyzoites present in cell culture supernatants were counted. Bars represent means of five biological replicates ± standard deviation.

**Figure 3 pathogens-10-00395-f003:**
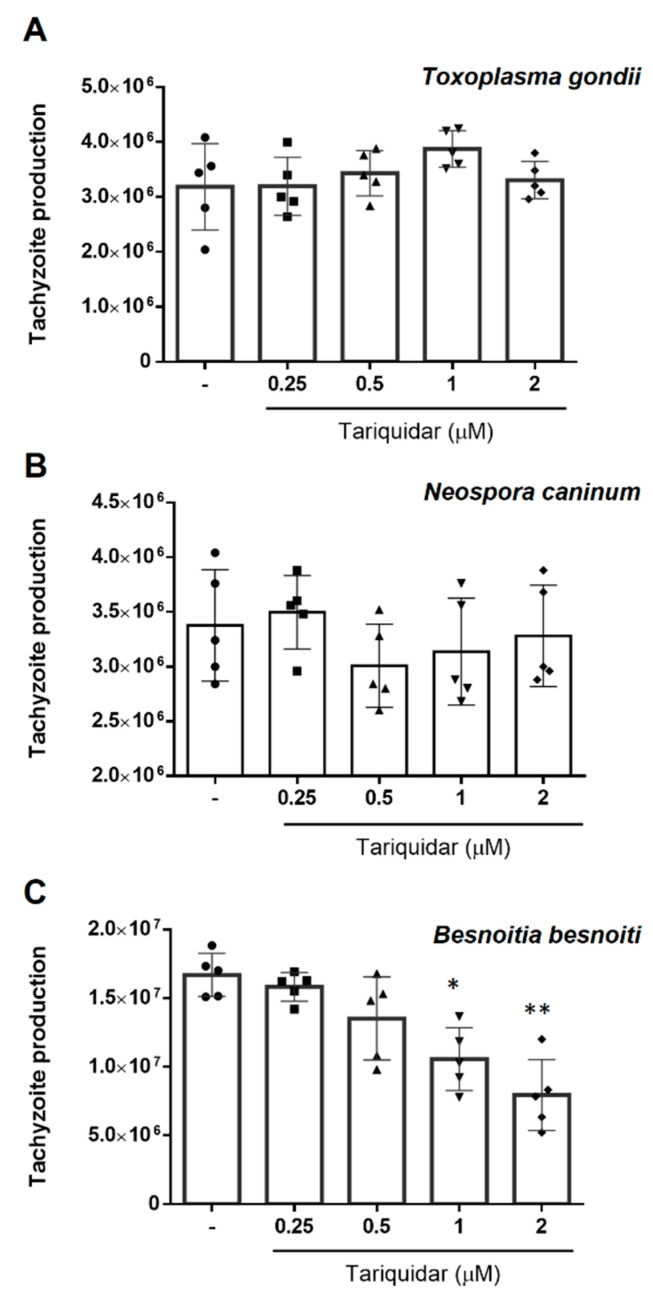
Tariquidar treatment differentially affects *T. gondii*, *N. caninum* and *B. besnoiti* tachyzoite production. The hosts cells pre-treated with tariquidar (0.25, 0.5 and 2 µM) were infected with *T. gondii* (**A**), *N. caninum* (**B**) or *B. besnoiti* (**C**) tachyzoites in inhibitor-free medium for 4 h, followed by the compound re-administration. At 48 h after infection, the number of tachyzoites present in cell culture supernatants were counted. Bars represent means of five biological replicates ± standard deviation.

**Figure 4 pathogens-10-00395-f004:**
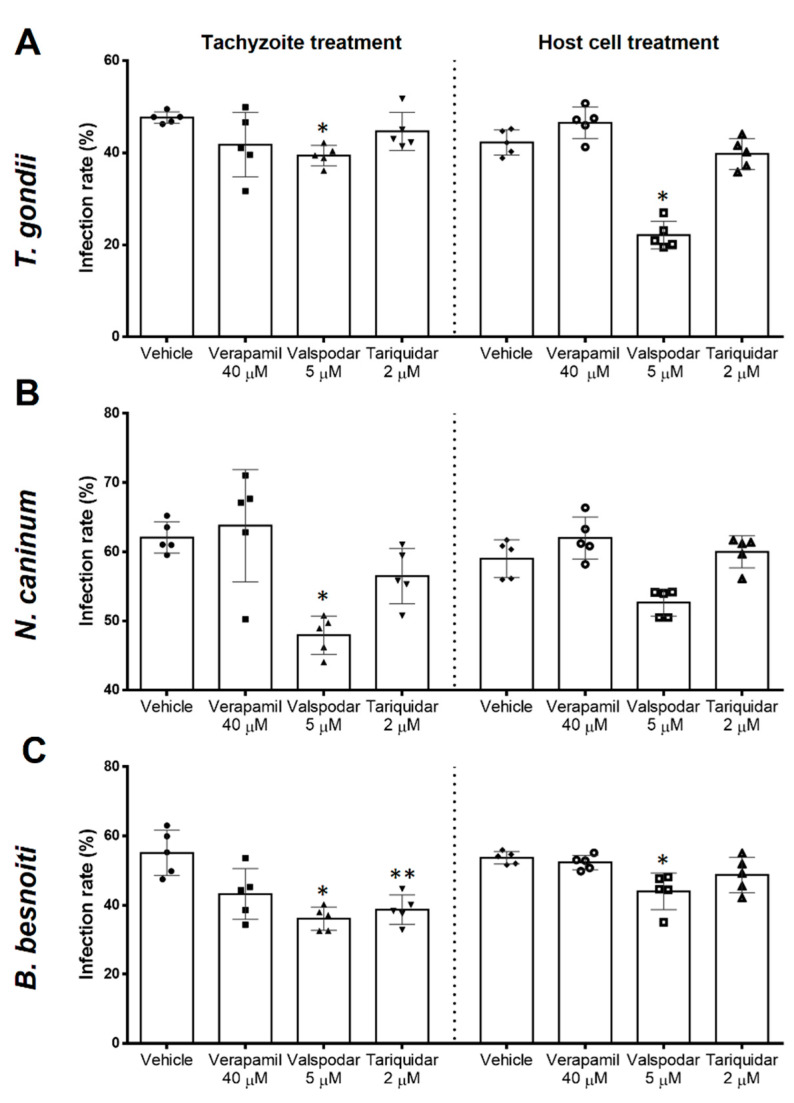
P-gp blockers differentially decrease *T. gondii*, *N. caninum* and *B. besnoiti* infection rates. Host cells or tachyzoites were pre-treated with verapamil (40 µM), valspodar (5 µM) or tariquidar (2 µM). Infection rates were determined 4 h p. i. with *T. gondii* (**A**), *N. caninum* (**B**) or *B. besnoiti* (**C**) tachyzoites. Bars represent means of five biological replicates ± standard deviation.

**Figure 5 pathogens-10-00395-f005:**
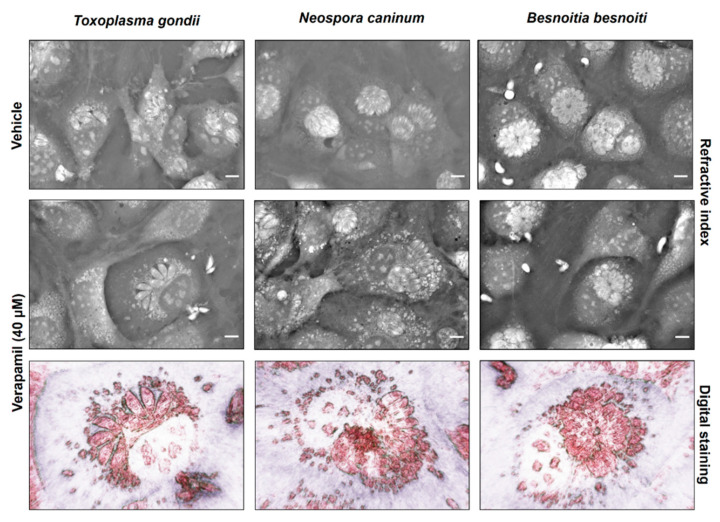
Live cell 3D-holotomography of *T. gondii*-, *N. caninum*- and *B. besnoiti*-infected and verapamil-treated BUVEC. Host cells were treated with verapamil (40 µM) and infected with *T. gondii*, *N. caninum* or *B. besnoiti* tachyzoites. Cell morphology was illustrated at 24 h p. i. via live 3D-holotomography. Images mirror refractive indices and used digital staining.

**Figure 6 pathogens-10-00395-f006:**
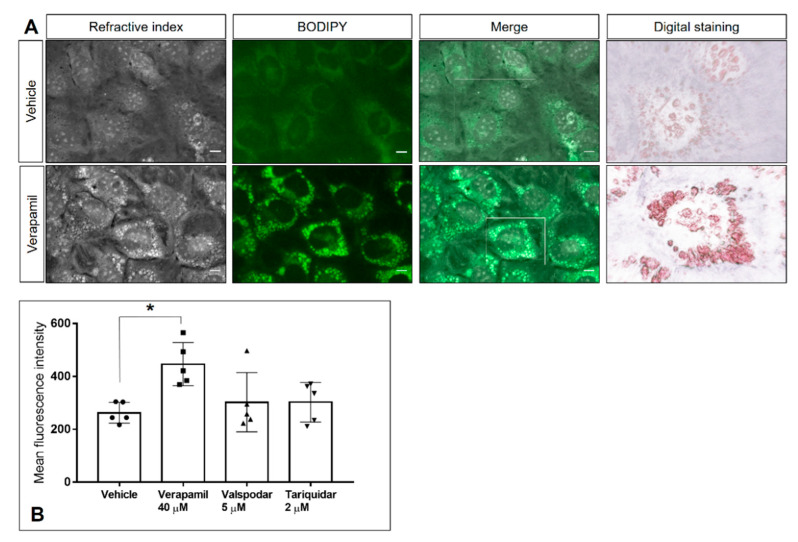
Lipid droplet accumulation induced by verapamil treatment. Lipid droplet formation was measured by FACS analysis in host cells 48 h after treatment with verapamil (40 µM), valspodar (5 µM) or tariquidar (2 µM) via Bodipy 493/503 staining. (**A**) Representative live cell 3D-holotomographic images of vehicle- or verapamil-treated BUVEC stained with Bodipy 493/503. The white frame indicates the area of digital staining, which was based on dense granule refractive indices. (**B**) Quantitative analysis of neutral lipid accumulation; graph bars represent the means of five biological replicates ± standard deviation.

**Figure 7 pathogens-10-00395-f007:**
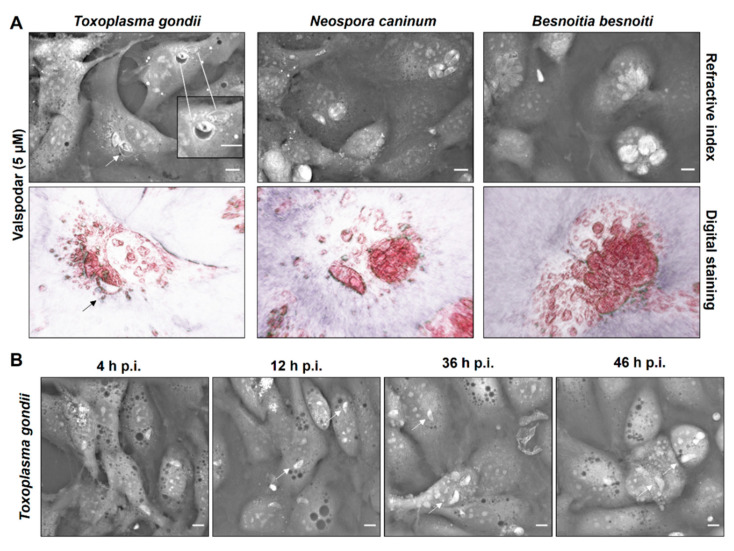
Live cell 3D-holotomography of *T. gondii*-, *N. caninum*- and *B. besnoiti*-infected and valspodar-treated BUVEC. (**A**) Host cells were treated with valspodar (5 µM) and infected with *T. gondii*, *N. caninum* or *B. besnoiti* tachyzoites. Cell morphology was illustrated at 24 h p. i. via live cell 3D-holotomography. Images mirror refractive indices (RI) and digital staining. Arrows point to the swollen PV in *T. gondii*-infected host cells treated with valspodar, in the zoomed image. (**B**) Time lapse of *T. gondii* development in valspodar (5 µM)-treated host cells illustrating the RI at 4, 12, 36 and 46 h p. i. Arrows point to non-dividing tachyzoites.

**Table 1 pathogens-10-00395-t001:** Summary of the effects of verapamil, valspodar and tariquidar on *T. gondii*, *N. caninum* and *B. besnoiti* as well as the host cells.

Effect	Verapamil	Valspodar	Tariquidar
Reduced tachyzoite proliferation	yes	yes	only *B. besnoiti*
Diminished meront size at 24 h p. i.	no	yes	no
Reduced infection rate	no	yes	only *B. besnoiti*
Decreased host cell permissiveness	no	only *T. gondii* and *B. besnoiti*	no
Neutral lipid accumulation	yes	no	no

## Data Availability

All data are included in the manuscript.

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
