# Peer review of "P-Glycoprotein Inhibitors Differently Affect Toxoplasma gondii, Neospora caninum and Besnoitia besnoiti Proliferation in Bovine Primary Endothelial Cells"

_pathogens, 2021, doi:10.3390/pathogens10040395_

Round 1
Reviewer 1 Report
This manuscript studied the effect of P-glycoprotein inhibitors on the effect of parasite proliferation and neutral lipid accumulation. The manuscript is prepared in high quality. Overall presentation of experimental details and data are exceptional. The study has scientific merit to demonstrate the effect of the P-glycoprotein inhibitors under the conditions the host cells were pretreated, or the parasite cell were pretreated with the P-glycoprotein inhibitors. To my understanding, the goal of the study is to determine the therapeutic potential of P-glycoprotein inhibitor drugs. However, the conditions tested in the paper does not necessarily reflect the real-world scenario that drug treatment is delivered post infection. I believe that it is important for the authors to test the effect of the P-glycoprotein inhibitors on the host cells that had been infected with parasites. That will add more significance to the current study.
Author Response
We appreciate your constructive suggestion. The potential effects of post-infection treatments on tachyzoite replication can indeed bring new insights into the application of these compounds in a clinical scenario. However, the main aim of this study was to establish the basics of potential PGP efficacy and to evaluate the differences of different PGP inhibitor generations in three different parasite species. Based on this study, the next step will be to screen effects in a post-infection scenario. Given that we here analyzed three different parasites at a time and that, consequently, such experiments require considerable labor and resources, we cannot include these data in the current study but will include this approach in the next studies.

Reviewer 2 Report
The manuscript by Larrazabal et al. represents a study on the effects of three P-glycoprotein inhibitors (verapamil, valspodar and tariquidar) on the replication and infectivity of tachyzoites of three species of cyst-forming coccidian (Toxoplasma gondii, Neospora caninum, and Besnoitia besnoiti), as well as on the neutral lipid content in host cells. The authors performed the analyses using the culture of bovine umbilical vein endotheliocytes and a combination of traditional light microscopy, fluorescence microscopy, 3D-holotomography, and flow cytometry. Their results demonstrate that the specificity and the range of observed effects vary among the three tested compounds. In general, the study appears to be reasonably designed and the obtained results are novel and scientifically sound. However, there are multiple flaws to be eliminated before publication of these results.
First of all, it is important to note the compositional issues. The lines 216-234 of the Discussion contain virtually the same information (review of literature), as can be found in the Introduction, although being better formulated. I suggest replacing the corresponding text in the introduction with this fragment. Then the Discussion can start form some introductory sentence followed by the actual discussion.
There are some experiments, which are mentioned in Material and Methods but not described in the Results, namely cytotoxicity assays and parasite viability tests. There is only a supplementary figure concerning this.
It remained unclear, why the authors used these particular ranges of concentrations for the three inhibitors. There is neither any reference to previous works nor their own tests justifying such a selection. In principle, that should be an extension to cytotoxicity assays, which have been performed here only with one concentration. This is not an idle question, since the authors discovered that tariquidar has only moderate effect on tachyzoite replication and only in a single species, Besnoitia besnoiti. As demonstrated for the two other inhibitors, the effect is strongly dosage-dependent. Therefore, higher tariquidar concentrations might affect also other species and demonstrate more pronounced effect(s) on B. besnoiti. The paper by Bottova et al., 2010, where a related compound (elacridar) showed effect against T. gondii at 10 um concentration, is in agreement with this hypothesis.
The authors' selection of what to show in figures is not always justified.
1) The Figure 5 is referenced in the text as the one, showing increased content of refractive granules (=lipid droplets) as a result of verapamil treatment. However, this figure shows only treated infected cells, whereas the untreated ones are omitted. It remains unclear, how an effect can be stated without a comparison. The authors should either add the missing panels, or remove the statement from the text. If they select the second option, then the Figure 5 becomes redundant, since it does not show anything important. Instead, I suggest replacing it with the Figure S2 (i.e. make it one of the major figures).
2) What is the reason to show (sub)panels with digital staining? Apart from the fact that they show zoomed portions of the main (sub)panels, they are nothing but redundant colorful pictures, bearing no additional information.
3) The Figure 7 demonstrates valspodar-driven suppression of meront development in the three coccidian species, resulting in the lower tachyzoite production. According to this figure, division arrest occurs only T. gondii. However, as judged by the Figure 2, in B. besnoiti, the suppression is comparable or even more dramatic (99,6% vs 98.9%). It is unclear, whether division arrest occurs also in this species. If no, then how this correlates with the result shown in the Figure 2? The text describing these results should be more comprehensive. Ideally, more (supplementary) photos for each species should be shown to demonstrate the variability of meront sizes. Moreover, I suggest a quantitative analysis: either comparison of meront size distributions, or distributions of the number of tachyzoites per meront.
The details of experiments on P-gp impact on tachyzoite replication are absolutely unclear. In the Material and Methods it is written, that there were two treatments of the same host cells: before infection and ~ 4 h post infection. Nothing is specified about this in the main text of the Results. However, in the captions to the Figures 1-3 it is written, that there was only a treatment for 48h before infection., i.e. a pre-treatment. This is further confusing, since there was another series of experiments, where pre-treatment was used to assess impact of the inhibitors on host cell permissiveness. Notably, nothing is mentioned concerning this series of experiments in Material and Methods!
The title reads ungrammatical. Probably, "efficacies" should be replaced by "effects". However, even after such a correction, it will not reflect all aspects of the study (see the very beginning of my comments, describing the manuscript contents. I suggest revise the title accordingly.
The text of the manuscript should be revised. There are grammatical and stylistic errors, redundant words as well as unclear/awkward sentences. Below are the suggested text changes line by line (note that after moving part of the text from the Discussion to the Introduction, some of these may become irrelevant):
Line 15: "auxotrophic of" > "auxotrophic for"
Lines 17-18 and elsewhere throughout the text: "We here analyzed" > either "Here, we analyzed…", or "We analyzed here".
Line 21: Nonetheless > in contrast. Alternatively the this sentence can be joined with the preceding one using "whereas" for junction.
Line 21: tariquidar treatments only affected B. besnoiti proliferation > tariquidar treatments affected proliferation only in B. besnoiti
Lines 21-23: repharase the sentence to make it simpler and clearer.
Line 23 and alsewhere: in case of > in the case of. By the way, its is
Line 24: infective capacities of > infection rates in
Lines 25-28: this awkward and confusing sentence needs a serious revision. The message here is absolutely unclear.
Lines 33-37: The order of taxonomical ranks is weird. In addition, due to ambiguous wording, it is unclear, what the authors meant by "large group", but apparently these are apicomplexans. If so, then the second sentence is wrong, since not all apicomplexans are intracellular. I suggest the following revision of these three sentences:
"Toxoplasma gondii, Neospora caninum and Besnoitia besnoiti are obligately intracellular parasitic protists known as cyst-forming coccidia and belonging to the family Sarcocystidae of the phylum Apicomplexa. Many apicomplexans are causal agents of important human and animal diseases with variable host specificity and clinical outcomes".
Reference to a 12-years-old review is not good. I suggest a more recent and relevant source: Votýpka J., Modrý D., Oborník M., Šlapeta J., Lukeš J. (2017) Apicomplexa. In: Archibald J., Simpson A., Slamovits C. (eds) Handbook of the Protists. Springer, Cham. https://doi.org/10.1007/978-3-319-28149-0_20
Line 38: T. gondii is NOT a major public and animal health problem. If you check the major problems at the WHO site, there will be no infectious diseases at all! In order to stress the importance of this species, the authors should mention that it has the widest known host range among eukaryotic parasites: virtually all mammals and even some birds.
Line 40: "has no zoonotic relevance" > does not infect humans
(of note, it does not have anthroponotic relevance as well)
Line 42: which is considered emergent in Europe leading to > which is emergent in Europe and leads to
Line 43: and to bull infertility > as well as infertility in bulls
Line 44: coccidian parasites > cyst-forming coccidia
(this sentence is not about all coccidia).
Line 45: remove the meaningless "thereby"
Line 46: delete "shortly after infection, and before conversion to bradyzoites." (this part makes absolutely no sense). Instead, the authors must properly explain the WHOLE development of cyst-forming coccidia, at least in the intermediate host. Otherwise, it is unclear, why exactly tachyzoite stage was selected for the experiments and why these coccidia are named "cyst-forming".
Line 47: bovine umbilical endothelial cells > bovine umbilical vein endothelial cells (BUVEC)
Line 49: delete "Moreover" and start a separate paragraph.
Line 53: kinds of cell types > types of host cells
Line 54: "and several parasites" > "T. gondii, N. caninum and B. besnoiti" (start a new sentence).
Line 55: " during acute replication" > "during acute stage of infection" or "during tachyzoite replication"
Line 56: lipid rich > lipid-rich
Line 57: for parasite replication > for parasite consumption (or uptake)
Lines 58-59: are reported to be significantly enhanced > become significantly increased
Lines 59-60: delete "In coccidian infections, the parasite depends on high amounts of cholesterol for tachyzoite multiplication but" This is the repeat of the information
Lines 60-61: faces the unique challenge > The parasites face the challenge…
Line 61: that will harm > that harm
Line 62: diminish > suppress are also regulated > are regulated
Line 64: molecules, which > molecules that
Line 65: one of the more > one of the most
Line 66: which is frequently correlated > the expression of which is frequently correlated
Line 71: metazoan parasites > helminths
Lines 73-74: delete "thereby reinforcing its pivotal role in parasite re-73 sistance development" (redundant)
Lines 75-76: "little is known on its physiological role in cell metabolism" > little is known about its impact on cell metabolism
Line 76: As such, P-gp seems involved > This protein seems to be involved
Line 78: (Bottova et al., 2009) > replace with [reference number]
Line 78: thereby suggesting that this protein might > and might
Line 79: delete "likewise"; two references do not correlate well with "several reports"
Lines 81-86: an awfully long and awkward sentence that must be replaced (the text in Discussion is definitely better).
Lines 87-88: of different P-gp inhibitor generations (i.e. verapamil, valspodar and tariquidar) > of verapamil, valspodar, and tariquidar, which represent the three generations of P-gp inhibitors.
Line 89: to tackle > to identify
Line 97: was diminished for > diminished by
Line 98: put comma before "respectively"
Line 100: to a diminishment > to a decrease
Line 101: of 36.6 ± 5.9 % > by 36.6 ± 5.9 %
Check also other instances of the same incorrect usage of prepositions before % in other experiments!
Line 106 (and in captions of other similar figures): replace "BUVEC" with "host cells"; abbreviations in captions are not good!
Line 107 (and in captions of other similar figures): delete "and thereafter infected"
Line 110 (an in other instances): treatments > treatment
Line 137: delete "here"
Line 138 (and elsewhere): vital > live
Line 140: delete "overall"
Lines 142-143: delete "a moderately reduced infective capacity of all three parasite species, thereby causing"
Line 143-145: "decrease of infection rates of 7.8 % (p = 0.009), 14.13 % (p = 0.013) and 19.3 % (p = 0.022) in case of T. gondii, N. caninum and B. besnoiti tachyzoites, respectively" > "decrease of infection rates with T. gondii, N. caninum, and B. besnoiti by 7.8 % (p = 0.009), 14.13 % (p = 0.013) and 19.3 % (p = 0.022), respectively"
Line 145: exclusively affected > affected exclusively
Line 148: impair > affect (or decrease)
Lines 152-153: host cell permissiveness > its permissiveness
Line 156: delete "host cells"
Line 157: as stated by > as judged by
Line 158: in all three parasites infections > with all three parasites
Line 161: delete either "host" or "primary bovine endothelial"
Lines 163-164: "Verapamil treatments", does this mean pre-treatments? It is irrelevant for uninfected cells, but important for infected ones. In Material and Methods it is written "as described above", but above it is totally messy (see my comments above)!
Lines 185-186: This last sentence is awkward and redundant. RI-based contrast is the essence of holotomography, nothing should be explained in addition. Mentioning software is also redundant here.
Line 192: "White square indicate digital staining area, which was performed according to" > "The white frame indicates the area of digital staining, which was performed based on"
Line 193: analysis on > analysis of
Line 194: bar graphs > graph bars
Line 197: delete "inhibitor treatment"
Line 198: stated > observed/documented/detected
Lines 198-199: "parasite-infected valspodar-treated BUVEC showed an impaired meront development for all parasite species" > "valspodar treatment led to impaired meront development in all parasite species"
Lines 202-203: appeared bloated/vacuolized and enlarged in size > appeared swollen
Lines 210-211: the same redundant sentence as in the previous figure caption.
Line 211: highlight PV vacuolization > point to swollen PVs
("PV vacuolization" is a tautology and makes no sense).
Lines 213-214: highlight the arrest in single tachyzoite stage > point to non-dividing tachyzoites
Line 217: fast replicating > fast-replicating
Line 217: which > that
Line 222: induce de novo synthesis > induce the de novo synthesis of this compound
Line 243 (and elsewhere): In line, > In line with this / In accordance with this
Line 251: other > additional
Line 253: glucose uptake [57] also in endothelial cells [58] > glucose uptake in various cell types, including endotheliocytes [57,58].
I suggest to leave here only the reference 58, as a more comprehensive one, and use the ref. #57 exclusievly in the next sentence (change the order of references accordingly!).
Line 253-255: "As such, verapamil blocked GLUT1-mediated glucose uptake in fibroblasts at both basal and stress-induced conditions [57]." > In fibroblasts, verapamil blocks GLUT1-mediated glucose transport at both basal and stress-induced conditions [57].
Lines 263-264: "developmental stagnation in all three parasites." This was demonstrated only for T. gondii. There are no illustrations on such effect in the other two species and this was not even mentioned in the Results.
Line 266: specie-specific > species-specific
Line 267: start a new sentence from "however"
Line 269: spell out "C. parvum", infected- > -infected (dash on a wrong side).
Lines 269-270: showing that valspodar-mediated efficacy seems conserved > suggesting that valspodar-mediated effects are conserved
Lines 270-273: Revise the sentence in the following way: "Given that valspodar is a derivative of cyclosporine, for which anti-parasitic activity against the same three apicomplexans has been demonstrated [59-61], these two compounds may have common effects.
Line 274: " three parasite species species which is in agreement to" > "three coccidian species studied here, which is in agreement with"
Line 276: thereby suggesting > suggesting (one thereby in a sentence is enough)
Line 281: proved evidence > evidence
Line 283: delete "residual"
Lines 286-287: was specifically developed for selectivity against P-gp > was developed as a specific P-gp inhibitor
Line 292-294: suggested revision: "…it has been previously reported that elacridar (10 μM) treatment reduced T. gondii proliferation and, by affecting Ca++ homeostasis of tachyzoites, led to hypermotility and untimely microneme secretion."
Line 298: an older report concludes > one report argues
Line 299: "cells [32,43] evidenced" > "cells, another evidenced" (move the references to the sentence end)
Lines 300-301: "However, the high variability of host cell types in the different studies might have contributed to" > "However, the fact that different studies were performed on different types of host cells can explain"
Line 303: delete "indeed" and start a new sentence
Line 308: failed to do so > did not have such an effect
Line 309: analyses on > analyses of
Line 310; actively be > be actively
Lines 310-311: BODIPY cannot be a P-gp substrate, it is a stain for neutral lipids!
Line 311: 493/505 > 493/503
Line 338: vital > live
Line 347: inhibitors were removed and after washing > the inhibitors were removed by washing
Line 348: plain medium > plain medium. (end of the sentence)
Line 348: cells were infected with > the cells were incubated with
Line 349: "a multiplicity of infection [36] of 1:5 for 4 h under inhibitor-free conditions." > "a multiplicity of infection of 5 for 4 h."
The reference provided here is absolutely irrelevant, the paper is about a non-infective disease. Multiplicity of infection is expressed by a single number (# of parasites / # of host cells), not a ratio. It s clear that the condition were inhibitor-free, since the inhibitor was removed by washing.
Line 354: specify the ranges of concentrations used in the work
Line 361: BUVEC were > the cells were
Line 362: induced an effect > exerted an effect
Line 363: inhibitor pre-treated cells > pre-treated cells
Line 368: assessed by FL-1 settings > assessed in the FL-1 channel
Line 373: CO2 (lower index)
Author Response
The manuscript by Larrazabal et al. represents a study on the effects of three P-glycoprotein inhibitors (verapamil, valspodar and tariquidar) on the replication and infectivity of tachyzoites of three species of cyst-forming coccidian (Toxoplasma gondii, Neospora caninum, and Besnoitia besnoiti), as well as on the neutral lipid content in host cells. The authors performed the analyses using the culture of bovine umbilical vein endotheliocytes and a combination of traditional light microscopy, fluorescence microscopy, 3D-holotomography, and flow cytometry. Their results demonstrate that the specificity and the range of observed effects vary among the three tested compounds. In general, the study appears to be reasonably designed and the obtained results are novel and scientifically sound. However, there are multiple flaws to be eliminated before publication of these results.
First of all, it is important to note the compositional issues. The lines 216-234 of the Discussion contain virtually the same information (review of literature), as can be found in the Introduction, although being better formulated. I suggest replacing the corresponding text in the introduction with this fragment. Then the Discussion can start form some introductory sentence followed by the actual discussion.
Response: We are thankful for the comment. As suggested, we changes the sentence from Discussion to Introduction (Lines 34-49). Additionally an introductory sentence was added into the discussion section (Line 224)
There are some experiments, which are mentioned in Material and Methods but not described in the Results, namely cytotoxicity assays and parasite viability tests. There is only a supplementary figure concerning this.
Response: We thank for the correction. The cytotoxicity data description were added in a new results section (Line 214)
It remained unclear, why the authors used these particular ranges of concentrations for the three inhibitors. There is neither any reference to previous works nor their own tests justifying such a selection. In principle, that should be an extension to cytotoxicity assays, which have been performed here only with one concentration. This is not an idle question, since the authors discovered that tariquidar has only moderate effect on tachyzoite replication and only in a single species, Besnoitia besnoiti. As demonstrated for the two other inhibitors, the effect is strongly dosage-dependent. Therefore, higher tariquidar concentrations might affect also other species and demonstrate more pronounced effect(s) on B. besnoiti. The paper by Bottova et al., 2010, where a related compound (elacridar) showed effect against T. gondii at 10 um concentration, is in agreement with this hypothesis.
Response: We really appreciate your observation regarding the P-gp inhibitors concentrations used in our work. Here we used verapamil (5 to 40 µM), valspodar (0.6 to 5 µM) and tariquidar (0.2 to 2 µM), according to concentrations in range of P-gp block from prior works:
- Tsuruo, T.; Iida, H.; Tsukagoshi, S.; Sakurai, Y. Overcoming of vincristine resistance in P388 leukemia in vivo and in vitro through enhanced cytotoxicity of vincristine and vinblastine by verapamil. Cancer research 1981, 41, 1967-1972
- Atadja, P.; Watanabe, T.; Xu, H.; Cohen, D. PSC-833, a frontier in modulation of P-glycoprotein mediated multidrug re-sistance. Cancer metastasis reviews 1998, 17, 163-168, doi:10.1023/a:1006046201497
In addition, we selected tariquidar as a highly selective P-gp inhibitor capable of inhibit P-gp with concentrations in nM concentration range:
- Roe, M.; Folkes, A.; Ashworth, P.; Brumwell, J.; Chima, L.; Hunjan, S.; Pretswell, I.; Dangerfield, W.; Ryder, H.; Charlton, P. Reversal of P-glycoprotein mediated multidrug resistance by novel anthranilamide derivatives. Bioorganic & medicinal chem-istry letters 1999, 9, 595-600, doi:10.1016/s0960-894x(99)00030-x
The use of a highly selective inhibitor of P-gp allowed us to discriminate potential P-gp independent effects mediated by verapamil or valspodar. Likewise, we agree that the increase of tariquidar concentration could evoke inhibitory effects on T. gondii and N. caninum tachyzoite proliferation in agreement of Bottova et al 2010. However, higher concentrations of tariquidar could also be a consequence of P-gp independent mechanisms associated to such concentrations, therefore we decided not to increase the concentrations.
The authors' selection of what to show in figures is not always justified.
1) The Figure 5 is referenced in the text as the one, showing increased content of refractive granules (=lipid droplets) as a result of verapamil treatment. However, this figure shows only treated infected cells, whereas the untreated ones are omitted. It remains unclear, how an effect can be stated without a comparison. The authors should either add the missing panels, or remove the statement from the text. If they select the second option, then the Figure 5 becomes redundant, since it does not show anything important. Instead, I suggest replacing it with the Figure S2 (i.e. make it one of the major figures).
Response: We appreciate the suggestion regarding the lack of untreated cells within the figure panel. We added infected cells untreated (vehicle) RI photos to the panel for comparison reasons. We left the Fig. S2 as it was.
2) What is the reason to show (sub)panels with digital staining? Apart from the fact that they show zoomed portions of the main (sub)panels, they are nothing but redundant colourful pictures, bearing no additional information.
Response: We are thankful for your comment regarding the usage of digital staining. This is the first publication that shows live 3D digital reconstruction of these three different cyst-forming coccidian together. Digital staining was performed using the R.I. of tachyzoites that showed to be common in all the parasites here studied (same settings applied to all the images showed). Additionally, digital staining permits an enhancement of the resolution in the zoomed areas, even beyond original resolution of the R.I.-based images, in order to clearly observe changes meronts throughout development.
3) The Figure 7 demonstrates valspodar-driven suppression of meront development in the three coccidian species, resulting in the lower tachyzoite production. According to this figure, division arrest occurs only T. gondii. However, as judged by the Figure 2, in B. besnoiti, the suppression is comparable or even more dramatic (99,6% vs 98.9%). It is unclear, whether division arrest occurs also in this species. If no, then how this correlates with the result shown in the Figure 2? The text describing these results should be more comprehensive. Ideally, more (supplementary) photos for each species should be shown to demonstrate the variability of meront sizes. Moreover, I suggest a quantitative analysis: either comparison of meront size distributions, or distributions of the number of tachyzoites per meront.
Response: The authors are thankful for this constructive comment. The inhibitory effect of valspodar at 5 uM in T. gondii and B. besnoiti at 48 h p.i. is similar, being the small differences observed non-significant and possibly driven by the data dispersion only. However, considering that the exemplary photo of B. besnoiti meront in Fig. 7 can be misleading (since we presented a cell that has multiple PVs), we changed the photo to a more representative one showing a diminished development of the meronts at 24 h. Additionally, the difference between valspodar-treated B. besnoiti meronts (Fig. 7) and non-treated condition is evident in the Fig 5 (vehicle). To avoid redundancy, we decided to included vehicle pictures once only.
As suggested by reviewer 2, an additional analysis of the meront size distribution was performed in valspodar-treated cells at 24 h p.i. by measurement of the meront length of at least 150 meronts per experimental condition (see Figure for Reviewer at the end of the manuscript). Overall, the effect of valspodar treatment reduced the meront length by 45.2 %, 40.2% and 29. 2% for T. gondii, N. caninum and B. besnoiti, confirming the impact in the tachyzoite development, we included this data in the result section (Line 199). We decided to include this data in the manuscript but not as figure.
4) The details of experiments on P-gp impact on tachyzoite replication are absolutely unclear. In the Material and Methods it is written, that there were two treatments of the same host cells: before infection and ~ 4 h post infection. Nothing is specified about this in the main text of the Results. However, in the captions to the Figures 1-3 it is written, that there was only a treatment for 48h before infection., i.e. a pre-treatment. This is further confusing, since there was another series of experiments, where pre-treatment was used to assess impact of the inhibitors on host cell permissiveness. Notably, nothing is mentioned concerning this series of experiments in Material and Methods!
Response: We appreciate this observation regarding confusing methods description. To avoid confusions, we incorporated additional details in the materials and methods section (Lines 345-363) and changed the caption of the figures to the following: “Verapamil treatments reduce T. gondii, N. caninum and B. besnoiti tachyzoite proliferation in a dose-dependent manner. Verapamil (5, 10, 20 and 40 µM) treated host cells with T. gondii (A), N. caninum (B) or B. besnoiti (C) tachyzoites in inhibitor-free medium for 4 h, followed by further treatment. At 48 h after infection, the number of tachyzoites present in cell culture supernatants were counted. Bars represent means of five biological replicates ± standard deviation.”
In detail, for proliferation inhibition experiments, host cells were treated for 48 h prior to infection. At that time, cells were washed to remove any traces of inhibitor and tachyzoites were added in inhibitor-free medium for 4 h to allow infection. After 4 h, monolayers were washed to remove any extracellular parasites and supplemented medium with each of the inhibitors and concentrations studied were again added to the infected monolayer for 48 h. At 4 h p.i., images for infection rate determination were done. As read-out, number of tachyzoites freshly released were determined at 48 h p.i.
For another experiment, tachyzoite stages were treated for 1 h, washed and allowed to infect host cell monolayers to study the ability of invasion of the treated tachyzoites versus non-treated parasites. We hope now it is clear how we performed the experiments.
The title reads ungrammatical. Probably, "efficacies" should be replaced by "effects". However, even after such a correction, it will not reflect all aspects of the study (see the very beginning of my comments, describing the manuscript contents. I suggest revise the title accordingly.
Response: We appreciate your recommendation regarding the title. We changed the manuscript title to: “P-glycoprotein inhibitors differentially affect Toxoplasma gondii, Neospora caninum and Besnoitia besnoiti proliferation in bovine primary endothelial cells”
The text of the manuscript should be revised. There are grammatical and stylistic errors, redundant words as well as unclear/awkward sentences. Below are the suggested text changes line by line (note that after moving part of the text from the Discussion to the Introduction, some of these may become irrelevant):
Line 15: "auxotrophic of" > "auxotrophic for"
Lines 17-18 and elsewhere throughout the text: "We here analyzed" > either "Here, we analyzed…", or "We analyzed here".
Line 21: Nonetheless > in contrast. Alternatively the this sentence can be joined with the preceding one using "whereas" for junction
Line 21: tariquidar treatments only affected B. besnoiti proliferation > tariquidar treatments affected proliferation only in B. besnoiti
Lines 21-23: repharase the sentence to make it simpler and clearer.
Line 23 and alsewhere: in case of > in the case of. By the way, its is
Line 24: infective capacities of > infection rates in
Lines 25-28: this awkward and confusing sentence needs a serious revision. The message here is absolutely unclear.
Lines 33-37: The order of taxonomical ranks is weird. In addition, due to ambiguous wording, it is unclear, what the authors meant by "large group", but apparently these are apicomplexans. If so, then the second sentence is wrong, since not all apicomplexans are intracellular. I suggest the following revision of these three sentences:
"Toxoplasma gondii, Neospora caninum and Besnoitia besnoiti are obligately intracellular parasitic protists known as cyst-forming coccidia and belonging to the family Sarcocystidae of the phylum Apicomplexa. Many apicomplexans are causal agents of important human and animal diseases with variable host specificity and clinical outcomes".
Reference to a 12-years-old review is not good. I suggest a more recent and relevant source: Votýpka J., Modrý D., Oborník M., Šlapeta J., Lukeš J. (2017) Apicomplexa. In: Archibald J., Simpson A., Slamovits C. (eds) Handbook of the Protists. Springer, Cham. https://doi.org/10.1007/978-3-319-28149-0_20 OK
Line 38: T. gondii is NOT a major public and animal health problem. If you check the major problems at the WHO site, there will be no infectious diseases at all! In order to stress the importance of this species, the authors should mention that it has the widest known host range among eukaryotic parasites: virtually all mammals and even some birds.
Line 40: "has no zoonotic relevance" > does not infect humans
(of note, it does not have anthroponotic relevance as well)
Line 42: which is considered emergent in Europe leading to > which is emergent in Europe and leads to
Line 43: and to bull infertility > as well as infertility in bulls
Line 44: coccidian parasites > cyst-forming coccidian
(this sentence is not about all coccidia).
Line 45: remove the meaningless "thereby"
Line 46: delete "shortly after infection, and before conversion to bradyzoites." (this part makes absolutely no sense). Instead, the authors must properly explain the WHOLE development of cyst-forming coccidia, at least in the intermediate host. Otherwise, it is unclear, why exactly tachyzoite stage was selected for the experiments and why these coccidia are named "cyst-forming".
Line 47: bovine umbilical endothelial cells > bovine umbilical vein endothelial cells (BUVEC)
Line 49: delete "Moreover" and start a separate paragraph.
Line 53: kinds of cell types > types of host cells
Line 54: "and several parasites" > "T. gondii, N. caninum and B. besnoiti" (start a new sentence).
Line 55: " during acute replication" > "during acute stage of infection" or "during tachyzoite replication"
Line 56: lipid rich > lipid-rich
Line 57: for parasite replication > for parasite consumption (or uptake)
Lines 58-59: are reported to be significantly enhanced > become significantly increased
Lines 59-60: delete "In coccidian infections, the parasite depends on high amounts of cholesterol for tachyzoite multiplication but" This is the repeat of the information
Lines 60-61: faces the unique challenge > The parasites face the challenge…
Line 61: that will harm > that harm
Line 62: diminish > suppress are also regulated > are regulated
Line 64: molecules, which > molecules that
Line 65: one of the more > one of the most
Line 66: which is frequently correlated > the expression of which is frequently correlated
Line 71: metazoan parasites > helminths
Lines 73-74: delete "thereby reinforcing its pivotal role in parasite re-73 sistance development" (redundant)
Lines 75-76: "little is known on its physiological role in cell metabolism" > little is known about its impact on cell metabolism
Line 76: As such, P-gp seems involved > This protein seems to be involved
Line 78: (Bottova et al., 2009) > replace with [reference number]
Line 78: thereby suggesting that this protein might > and might
Line 79: delete "likewise"; two references do not correlate well with "several reports"
Lines 81-86: an awfully long and awkward sentence that must be replaced (the text in Discussion is definitely better).
Lines 87-88: of different P-gp inhibitor generations (i.e. verapamil, valspodar and tariquidar) > of verapamil, valspodar, and tariquidar, which represent the three generations of P-gp inhibitors.
Line 89: to tackle > to identify
Line 97: was diminished for > diminished by
Line 98: put comma before "respectively"
Line 100: to a diminishment > to a decrease
Line 101: of 36.6 ± 5.9 % > by 36.6 ± 5.9 %
Check also other instances of the same incorrect usage of prepositions before % in other experiments!
Line 106 (and in captions of other similar figures): replace "BUVEC" with "host cells"; abbreviations in captions are not good!
Line 107 (and in captions of other similar figures): delete "and thereafter infected"
Line 110 (an in other instances): treatments > treatment
Line 137: delete "here"
Line 138 (and elsewhere): vital > live
Line 140: delete "overall"
Lines 142-143: delete "a moderately reduced infective capacity of all three parasite species, thereby causing"
Line 143-145: "decrease of infection rates of 7.8 % (p = 0.009), 14.13 % (p = 0.013) and 19.3 % (p = 0.022) in case of T. gondii, N. caninum and B. besnoiti tachyzoites, respectively" > "decrease of infection rates with T. gondii, N. caninum, and B. besnoiti by 7.8 % (p = 0.009), 14.13 % (p = 0.013) and 19.3 % (p = 0.022), respectively"
Line 145: exclusively affected > affected exclusively
Line 148: impair > affect (or decrease)
Lines 152-153: host cell permissiveness > its permissiveness
Line 156: delete "host cells"
Line 157: as stated by > as judged by
Line 158: in all three parasites infections > with all three parasites
Line 161: delete either "host" or "primary bovine endothelial"
Lines 163-164: "Verapamil treatments", does this mean pre-treatments? It is irrelevant for uninfected cells, but important for infected ones. In Material and Methods it is written "as described above", but above it is totally messy (see my comments above)!
Lines 185-186: This last sentence is awkward and redundant. RI-based contrast is the essence of holotomography, nothing should be explained in addition. Mentioning software is also redundant here.
Line 192: "White square indicate digital staining area, which was performed according to" > "The white frame indicates the area of digital staining, which was performed based on"
Line 193: analysis on > analysis of
Line 194: bar graphs > graph bars
Line 197: delete "inhibitor treatment"
Line 198: stated > observed/documented/detected
Lines 198-199: "parasite-infected valspodar-treated BUVEC showed an impaired meront development for all parasite species" > "valspodar treatment led to impaired meront development in all parasite species"
Lines 202-203: appeared bloated/vacuolized and enlarged in size > appeared swollen
Lines 210-211: the same redundant sentence as in the previous figure caption.
Line 211: highlight PV vacuolization > point to swollen PVs
("PV vacuolization" is a tautology and makes no sense).
Lines 213-214: highlight the arrest in single tachyzoite stage > point to non-dividing tachyzoites
Line 217: fast replicating > fast-replicating
Line 217: which > that
Line 222: induce de novo synthesis > induce the de novo synthesis of this compound
Line 243 (and elsewhere): In line, > In line with this / In accordance with this
Line 251: other > additional
Line 253: glucose uptake [57] also in endothelial cells [58] > glucose uptake in various cell types, including endotheliocytes [57,58].
I suggest to leave here only the reference 58, as a more comprehensive one, and use the ref. #57 exclusievly in the next sentence (change the order of references accordingly!).
Line 253-255: "As such, verapamil blocked GLUT1-mediated glucose uptake in fibroblasts at both basal and stress-induced conditions [57]." > In fibroblasts, verapamil blocks GLUT1-mediated glucose transport at both basal and stress-induced conditions [57].
Lines 263-264: "developmental stagnation in all three parasites." This was demonstrated only for T. gondii. There are no illustrations on such effect in the other two species and this was not even mentioned in the Results.
Line 266: specie-specific > species-specific
Line 267: start a new sentence from "however"
Line 269: spell out "C. parvum", infected- > -infected (dash on a wrong side).
Lines 269-270: showing that valspodar-mediated efficacy seems conserved > suggesting that valspodar-mediated effects are conserved
Lines 270-273: Revise the sentence in the following way: "Given that valspodar is a derivative of cyclosporine, for which anti-parasitic activity against the same three apicomplexans has been demonstrated [59-61], these two compounds may have common effects.
Line 274: " three parasite species species which is in agreement to" > "three coccidian species studied here, which is in agreement with"
Line 276: thereby suggesting > suggesting (one thereby in a sentence is enough)
Line 281: proved evidence > evidence
Line 283: delete "residual"
Lines 286-287: was specifically developed for selectivity against P-gp > was developed as a specific P-gp inhibitor
Line 292-294: suggested revision: "…it has been previously reported that elacridar (10 μM) treatment reduced T. gondii proliferation and, by affecting Ca++ homeostasis of tachyzoites, led to hypermotility and untimely microneme secretion."
Line 298: an older report concludes > one report argues
Line 299: "cells [32,43] evidenced" > "cells, another evidenced" (move the references to the sentence end)
Lines 300-301: "However, the high variability of host cell types in the different studies might have contributed to" > "However, the fact that different studies were performed on different types of host cells can explain"
Line 303: delete "indeed" and start a new sentence
Line 308: failed to do so > did not have such an effect
Line 309: analyses on > analyses of
Line 310; actively be > be actively
Lines 310-311: BODIPY cannot be a P-gp substrate, it is a stain for neutral lipids!
Line 311: 493/505 > 493/503
Line 338: vital > live
Line 347: inhibitors were removed and after washing > the inhibitors were removed by washing
Line 348: plain medium > plain medium. (end of the sentence)
Line 348: cells were infected with > the cells were incubated with
Line 349: "a multiplicity of infection [36] of 1:5 for 4 h under inhibitor-free conditions." > "a multiplicity of infection of 5 for 4 h."
The reference provided here is absolutely irrelevant, the paper is about a non-infective disease. Multiplicity of infection is expressed by a single number (# of parasites / # of host cells), not a ratio. It s clear that the condition were inhibitor-free, since the inhibitor was removed by washing.
Line 354: specify the ranges of concentrations used in the work
Line 361: BUVEC were > the cells were
Line 362: induced an effect > exerted an effect
Line 363: inhibitor pre-treated cells > pre-treated cells
Line 368: assessed by FL-1 settings > assessed in the FL-1 channel
Line 373: CO2 (lower index)
Response: We are truly thankful for all the detailed observations and the correction of the mistakes present in the manuscript. The specific comments were added or corrected in the present version of the manuscript as suggested.

Reviewer 3 Report
The present study compares different generations of p-gp inhibitors in three coccidian species of big impact in veterinary and/or human health.
Introduction is well written and documented and the objectives of the study are very clear: to compare the effects of the different inhibitors in the 3 species.
Research was performed accordingly in a correct way, nevertheless some minor issues must be adressed before acceptance of the manuscript.
First, some acronyms, as p-gp, must be included in the introduction, even though they are in the abstract. The most concerning issue is related to discussion, the potential toxic effect of the onhibitors in the host cells has been included in the supplementary material which is correct, but it is important to raise the results of that assay during result and discussion sections, in order to make conclusions letting the reader known that toxicity was taken into account to get such conclusions.
Moreover, to hyphothesize about calcium or glucose influence is good, to put in context the results and conclusions obtained but conclusions about calcium per example cannot be taken into consideration here only based on bibliographic references. Discussion may need some rewriting.
Finally, I would like to suggest the authors to include a summary table with the comparison of the 3 drugs in the 3 species including the results obtained to help visualize and compare them all.
Author Response
The present study compares different generations of p-gp inhibitors in three coccidian species of big impact in veterinary and/or human health. Introduction is well written and documented and the objectives of the study are very clear: to compare the effects of the different inhibitors in the 3 species. Research was performed accordingly in a correct way, nevertheless some minor issues must be addressed before acceptance of the manuscript:
First, some acronyms, as p-gp, must be included in the introduction, even though they are in the abstract.
Response: We are thankful from the comment. To improve the quality of the manuscript we added the missing acronyms in the introduction (line 70)
The most concerning issue is related to discussion, the potential toxic effect of the inhibitors in the host cells has been included in the supplementary material which is correct, but it is important to raise the results of that assay during result and discussion sections, in order to make conclusions letting the reader known that toxicity was taken into account to get such conclusions.
Response: We appreciate the observation and we added this in a new results section (line 214) and in the discussion of each compound (verapamil line 232, Valspodar line 259 and tariquidar line 287).
Moreover, to hypothesize about calcium or glucose influence is good, to put in context the results and conclusions obtained but conclusions about calcium per example cannot be taken into consideration here only based on bibliographic references. Discussion may need some rewriting.
Response: The authors are thankful for the recommendation. The suggested changes were included. First, in the line 254 was added “however further experiments are necessary to address this possibility” to avoid misleading conclusions. Additionally the calcium part were rephrased (line 293) for the same reason “…it has been previously reported that elacridar (10 μM) treatment reduced T. gondii proliferation and, by affecting Ca++ homeostasis of tachyzoites, led to hypermotility and untimely microneme secretion”, also suggested by Reviewer 2.
Finally, I would like to suggest the authors to include a summary table with the comparison of the 3 drugs in the 3 species including the results obtained to help visualize and compare them all.
Response: We are very thankful for the suggestion. We agree that due to the amount of experimental conditions the table will certainly improve the manuscript quality and will make it easier for the reader to have an overview on our results. The summary table was added to the new version of the manuscript (Line 324).

Round 2
Reviewer 1 Report
Thank you for addressing the comments. The manuscript will benefit the audience who is seeking information on the effect of P-glycoprotein inhibitors.
Author Response
Dear reviewer 1:
We are thankful for the comments.
Reviewer 2 Report
After the revision made by the authors some issues still remain and need to be solved.
I appreciate that the authors performed the analysis of the meront size for treated and untreated condition in the three studied species, but disagree with hiding the data. This is unacceptable since it does not meet the requirements of a reproducible research, especially given that it is stated that " All data are included in the manuscript". I insist on including the missing figure into the manuscript (either as a main or as a supplementary). In addition, this analysis must be complemented with the test for the statistical significance of the difference in the mean values.
It is good that the Figure 5 has been updated by adding the photos of untreated host cells. However, now the interpretation of these data seems to be overstated. The difference in the abundance of globular structures between the two conditions can be seen only in the case of Neospora caninum, whereas for the two other species such differences are not detectable! Meanwhile, quantitative data are missing. Therefore, the corresponding sentence should be rewritten. In addition, as can be seen from the Suppl. Figure 2, the accumulation of the globular structures is associated with a different distribution within the cell: they move from the center to the periphery of the cell.
The suggested version:
"In uninfected BUVEC cells, the 48 h verapamil treatment induced a considerable accumulation of the dense globular structures [refractive index of 1.3488 ± 0.0048] in the cytoplasm and their displacement from the center to the periphery of the cells (Supplementary Fig. 2). As for infected cells, this effect could be clearly observed only in the case of N. caninum, whereas for the two other coccidian specis the results were inconclusive (Fig. 5)."
In the text, the authors ignored some mistakes I had previously mentioned and introduced new ones. It was not so easy to track them especially given that the version with tracking changes does not really reflect all the differences between the old and the new version. Below is the list of what is to be corrected with indication of line numbers:
13 (AND 6 MORE INSTANCES THROUGHOUT THE TEXT): In case > in the case.
Only one or two instance have been previously corrected, but they should be fixed all!!! Use "find and replace" function!
16: transporter is involved > transporter involved
25-28: I have previously asked to revise this sentence. Suggested version: "The absence of pronounced anti-parasitic impact of tariquidar, which represents here the most selective P-gp inhibitor, suggests that the observed effects of verapamil and valspodar are associated with mechanisms independent of P-gp."
28-29: the sentence is misleading. Only in the case of tariquidar there is a significant difference between B. besnoiti and the other species. Suggested version: "Out of the three species tested here, this compound affected only B. besnoiti proliferation and its effect was much milder as compared to verapamil and valspodar".
39-40: Human and animal health are affected > This species represents a serious health threat
42: this is the second sentence starting with "in contrast". Just remove it from here, or replace with "The third species,".
43: of skin > of the skin
44: remove the redundant "Furthermore"
46: start a new sentence from "this stage"
48: bradizoites > bradyzoites
48: encapsulated within > enclosed in
48: which are the infective stage for > which are infective for
49: add "(felids and canids)" after "hosts"
50: along > in
58: remove double space
60-63: this sentence should be simplified as follows: "To prevent toxic accumulation of free cholesterol in the cell, most excess of it is esterified and stored in lipid-rich organelles, such as lipid droplets [18,19], being available for parasite consumption [8,13]."
64: sizes and numbers > size and number
65: enhanced > increased
66: efflux via > use of
70: member > members
74: participation > the participation
74: "not only" > "also" (because this comes in addition to the cancer cells)
75: or > and
76: but also > as well as
77: or > and
77: well-known > the well known
81-82: This sentence is awkward and partially false: the second reference is not for a coccidian. The suggested version: "The P-gp inhibition was demonstrated to suppress replication in some coccidia and microsporidia [31,32]."
83: why "several"? How does this correlate with the text below that "the three inhibitors represented each P-gp blocker generation"? In the previous version, there were only three generations. Replace back "several" by "three".
86: efficacies > effects
91: to compare and to identify > to identify and compare
97: in medium > in the medium
104: here studied > studied here
106: reduce > reduces
107: Verapamil (5, 10, 20 and 40 μM)treated host cells > The hosts cells pre-treated with verapamil (5, 10, 20 and 40 μM)
108: in > in the
109: by further treatment > by the compound re-administration
NOTE: do the same above edits for the legends of the Figures 2 and 3.
148: in > by
160: of > by
168: globular > the globular
180: intensities > intensity
197: here tested > tested here
198: here studied > studied here
200: length > diameter (Is it possible to identify length and width in nearly round objects?)
201: "(data not shown)" > put here the reference to the figure that you must add
211 swollen > the swollen
211: in zoom image > in the zoomed image
214: "P-gp inhibitor treatment do not produce cytotoxic damage to host cells nor tachyzoites" > "P-gp inhibitor treatment does not cause cytotoxic damage to host cells or tachyzoites"
215: cytotoxic effect > a cytotoxic effect
219: trypan > the trypan
224: coccidian parasites here studied > coccidia studied here
226-231: revise the text as follows: "Being auxotrophic for cholesterol, they strongly depend on the availability and efficient uptake of this nutrient for the successful proliferation [39]. Hence, P-gp implicated in the cholesterol uptake represents a promising anti-coccidial target. Here we investigated the effects of different P-gp inhibitors on the three above-mentioned parasite species."
231: With the above edits, the sentence should start with "We found that…"
232: remove comma
233: delete "treatment"
234: at > by
235-237: revise the sentence as follows: "Comparable effects of verapamil (at concentrations of 10-100 μM) were already reported in T. gondii infections of mouse embryonic fibroblasts and enterocytes applying [40,41]."
238: We here additionally demonstrated > Here we demonstrated
243: anti-coccidian > anti-coccidial
245-246: was previously linked to > has been previously demonstrated to depend on
247: it seems likely to assume that > it is likely that
256: delete "furthermore"
260-261: Thus, 5 μM valspodar reduced 97 ± 3.7% of tachyzoite production of all coccidian species here studied," > "Thus, 5 μM valspodar depending on the studied species reduced tachyzoite production by 92.8 ± 2.5 – 99.6 ± 0.3 %,"
261: concentration eight times smaller > concentration being eight times smaller
262: in case of verapamil > that of verapamil
263: delete "and reduced developmentin all three parasites". What is observed, is meront size reduction. The specifics of T. gondii are explained in the next sentence.
263-265: Suggested sentence revision: "In T. gondii, 3D-holotomographic microscopy also revealed an arrest at the single tachyzoite-stage."
266: put period between the sentences
268: C. parvum-infected > Cryptosporidium parvum-infected
(this is the first mentioning of this species in the manuscript)
269: apicomplexan species > these apicomplexans
271: the same three apicomplexans > the same three species
272: We here > Here we
273: in three > in the three
275-276: pre-treatment led > pre-treatment with this compound led
276: significant > moderate
277: comma before "thereby"
278: comma before "even"
279: impaired > impairs
284: delete "here"
285-286: Revise the sentence as follows: "Tariquidar is a potent P-gp-specific allosteric inhibitor with an average IC50 of 50 nM [28,37,38]".
288: showed significant efficacy > showed a moderate effect
290: comma before "thereby"
292: In line to > In line with
302: P-pg > P-gp
304: to address if > to find out if
308: comma before "whilst"
313: as inducer > as an inducer
316: as is showed in the summary table 1 > as shown in the Table 1
316: we here demonstrated > here we demoinstrated
318: showed > possess
319: Overall, we here assume > We assume
320-321: and suggesting independentce of > and is independent of
323: delete "case of"
325: proliferation in host cells > as well as the host cells
Table 1: Meront development diminuished > diminished meront size
Table 1: Tachyzoite infection rate impairment > reduced infection rate
330: where > were
351-352: After a 48 h of treatment" > After 48 h of treatment
363: Put back the information about the experiments on the host cells' permissiveness!
395: and samples > and the samples
407: arithmetic > the arithmetic
408: comparisons > the comparison
409: Kruskal-Wallis tests were performed > the Kruskal-Wallis test was used
410: the global comparisons by > a global comparison by the
411: tests indicated > test indicated
411-412: multiple comparison tests were carried out via Dunn tests > multiple comparison was carried out using the Dunn test
518: italicize Encephalitozoon
Author Response
After the revision made by the authors some issues still remain and need to be solved.
I appreciate that the authors performed the analysis of the meront size for treated and untreated condition in the three studied species, but disagree with hiding the data. This is unacceptable since it does not meet the requirements of a reproducible research, especially given that it is stated that " All data are included in the manuscript". I insist on including the missing figure into the manuscript (either as a main or as a supplementary). In addition, this analysis must be complemented with the test for the statistical significance of the difference in the mean values.
Answer: We are thankful for this constructive observation. The new data regarding the effect of valspodar on meront development was included in this version of the manuscript as supplementary figure 4 (attached after the references), while the statistical analysis is included in the line 206 from the new manuscript version.
It is good that the Figure 5 has been updated by adding the photos of untreated host cells. However, now the interpretation of these data seems to be overstated. The difference in the abundance of globular structures between the two conditions can be seen only in the case of Neospora caninum, whereas for the two other species such differences are not detectable! Meanwhile, quantitative data are missing. Therefore, the corresponding sentence should be rewritten. In addition, as can be seen from the Suppl. Figure 2, the accumulation of the globular structures is associated with a different distribution within the cell: they move from the center to the periphery of the cell.
The suggested version:
"In uninfected BUVEC cells, the 48 h verapamil treatment induced a considerable accumulation of the dense globular structures [refractive index of 1.3488 ± 0.0048] in the cytoplasm and their displacement from the center to the periphery of the cells (Supplementary Fig. 2). As for infected cells, this effect could be clearly observed only in the case of N. caninum, whereas for the two other coccidian species the results were inconclusive (Fig. 5)."
Answer: We appreciate the comment regarding this result. The corrections were added as suggested in the new manuscript version (line 166-171)
In the text, the authors ignored some mistakes I had previously mentioned and introduced new ones. It was not so easy to track them especially given that the version with tracking changes does not really reflect all the differences between the old and the new version. Below is the list of what is to be corrected with indication of line numbers:
13 (AND 6 MORE INSTANCES THROUGHOUT THE TEXT): In case > in the case.
Only one or two instance have been previously corrected, but they should be fixed all!!! Use "find and replace" function!
16: transporter is involved > transporter involved
25-28: I have previously asked to revise this sentence. Suggested version: "The absence of pronounced anti-parasitic impact of tariquidar, which represents here the most selective P-gp inhibitor, suggests that the observed effects of verapamil and valspodar are associated with mechanisms independent of P-gp."
28-29: the sentence is misleading. Only in the case of tariquidar there is a significant difference between B. besnoiti and the other species. Suggested version: "Out of the three species tested here, this compound affected only B. besnoiti proliferation and its effect was much milder as compared to verapamil and valspodar".
39-40: Human and animal health are affected > This species represents a serious health threat
42: this is the second sentence starting with "in contrast". Just remove it from here, or replace with "The third species,".
43: of skin > of the skin
44: remove the redundant "Furthermore"
46: start a new sentence from "this stage"
48: bradizoites > bradyzoites
48: encapsulated within > enclosed in
48: which are the infective stage for > which are infective for
49: add "(felids and canids)" after "hosts"
50: along > in
58: remove double space
60-63: this sentence should be simplified as follows: "To prevent toxic accumulation of free cholesterol in the cell, most excess of it is esterified and stored in lipid-rich organelles, such as lipid droplets [18,19], being available for parasite consumption [8,13]."
64: sizes and numbers > size and number
65: enhanced > increased
66: efflux via > use of
70: member > members
74: participation > the participation
74: "not only" > "also" (because this comes in addition to the cancer cells)
75: or > and
76: but also > as well as
77: or > and
77: well-known > the well known
81-82: This sentence is awkward and partially false: the second reference is not for a coccidian. The suggested version: "The P-gp inhibition was demonstrated to suppress replication in some coccidia and microsporidia [31,32]."
83: why "several"? How does this correlate with the text below that "the three inhibitors represented each P-gp blocker generation"? In the previous version, there were only three generations. Replace back "several" by "three".
86: efficacies > effects
91: to compare and to identify > to identify and compare
97: in medium > in the medium
104: here studied > studied here
106: reduce > reduces
107: Verapamil (5, 10, 20 and 40 μM)treated host cells > The hosts cells pre-treated with verapamil (5, 10, 20 and 40 μM)
108: in > in the
109: by further treatment > by the compound re-administration
NOTE: do the same above edits for the legends of the Figures 2 and 3.
148: in > by
160: of > by
168: globular > the globular
180: intensities > intensity
197: here tested > tested here
198: here studied > studied here
200: length > diameter (Is it possible to identify length and width in nearly round objects?)
201: "(data not shown)" > put here the reference to the figure that you must add
211 swollen > the swollen
211: in zoom image > in the zoomed image
214: "P-gp inhibitor treatment do not produce cytotoxic damage to host cells nor tachyzoites" > "P-gp inhibitor treatment does not cause cytotoxic damage to host cells or tachyzoites"
215: cytotoxic effect > a cytotoxic effect
219: trypan > the trypan
224: coccidian parasites here studied > coccidia studied here
226-231: revise the text as follows: "Being auxotrophic for cholesterol, they strongly depend on the availability and efficient uptake of this nutrient for the successful proliferation [39]. Hence, P-gp implicated in the cholesterol uptake represents a promising anti-coccidial target. Here we investigated the effects of different P-gp inhibitors on the three above-mentioned parasite species."
231: With the above edits, the sentence should start with "We found that…"
232: remove comma
233: delete "treatment"
234: at > by
235-237: revise the sentence as follows: "Comparable effects of verapamil (at concentrations of 10-100 μM) were already reported in T. gondii infections of mouse embryonic fibroblasts and enterocytes applying [40,41]."
238: We here additionally demonstrated > Here we demonstrated
243: anti-coccidian > anti-coccidial
245-246: was previously linked to > has been previously demonstrated to depend on
247: it seems likely to assume that > it is likely that
256: delete "furthermore"
260-261: Thus, 5 μM valspodar reduced 97 ± 3.7% of tachyzoite production of all coccidian species here studied," > "Thus, 5 μM valspodar depending on the studied species reduced tachyzoite production by 92.8 ± 2.5 – 99.6 ± 0.3 %,"
261: concentration eight times smaller > concentration being eight times smaller
262: in case of verapamil > that of verapamil
263: delete "and reduced developmentin all three parasites". What is observed, is meront size reduction. The specifics of T. gondii are explained in the next sentence.
263-265: Suggested sentence revision: "In T. gondii, 3D-holotomographic microscopy also revealed an arrest at the single tachyzoite-stage."
266: put period between the sentences
268: C. parvum-infected > Cryptosporidium parvum-infected
(this is the first mentioning of this species in the manuscript)
269: apicomplexan species > these apicomplexans
271: the same three apicomplexans > the same three species
272: We here > Here we
273: in three > in the three
275-276: pre-treatment led > pre-treatment with this compound led
276: significant > moderate
277: comma before "thereby"
278: comma before "even"
279: impaired > impairs
284: delete "here"
285-286: Revise the sentence as follows: "Tariquidar is a potent P-gp-specific allosteric inhibitor with an average IC50 of 50 nM [28,37,38]".
288: showed significant efficacy > showed a moderate effect
290: comma before "thereby"
292: In line to > In line with
302: P-pg > P-gp
304: to address if > to find out if
308: comma before "whilst"
313: as inducer > as an inducer
316: as is showed in the summary table 1 > as shown in the Table 1
316: we here demonstrated > here we demoinstrated
318: showed > possess
319: Overall, we here assume > We assume
320-321: and suggesting independentce of > and is independent of
323: delete "case of"
325: proliferation in host cells > as well as the host cells
Table 1: Meront development diminuished > diminished meront size
Table 1: Tachyzoite infection rate impairment > reduced infection rate
330: where > were
351-352: After a 48 h of treatment" > After 48 h of treatment
363: Put back the information about the experiments on the host cells' permissiveness!
395: and samples > and the samples
407: arithmetic > the arithmetic
408: comparisons > the comparison
409: Kruskal-Wallis tests were performed > the Kruskal-Wallis test was used
410: the global comparisons by > a global comparison by the
411: tests indicated > test indicated
411-412: multiple comparison tests were carried out via Dunn tests > multiple comparison was carried out using the Dunn test
518: italicize Encephalitozoon
Answer: we truly appreciate these observations regarding grammar along the text. All the corrections were performed in the new version of the manuscript.
